# Neural Regression, Representational Similarity, Model Zoology & Neural Taskonomy at Scale in Rodent Visual Cortex

**Colin Conwell**[*]
Department of Psychology
Harvard University

**David Mayo**
CSAIL & CBMM
MIT

**Michael A. Buice**
Modeling, Analysis & Theory
Allen Institute MindScope Program

**Boris Katz**
CSAIL & CBMM
MIT

**George A. Alvarez**
Department of Psychology
Harvard University

**Andrei Barbu**
CSAIL & CBMM
MIT

## Abstract

How well do deep neural networks fare as models of mouse visual cortex? A majority of research to date suggests results far more mixed than those produced in the modeling of primate visual cortex. Here, we perform a large-scale benchmarking of dozens of deep neural network models in mouse visual cortex with both representational similarity analysis and neural regression. Using the Allen Brain Observatory's 2-photon calcium-imaging dataset of activity in over 6,000 reliable rodent visual cortical neurons recorded in response to natural scenes, we replicate previous findings and resolve previous discrepancies, ultimately demonstrating that modern neural networks can in fact be used to explain activity in the mouse visual cortex to a more reasonable degree than previously suggested. Using our benchmark as an atlas, we offer preliminary answers to overarching questions about levels of analysis (e.g. do models that better predict the representations of individual neurons also predict representational similarity across neural populations?); questions about the properties of models that best predict the visual system overall (e.g. is convolution or category-supervision necessary to better predict neural activity?); and questions about the mapping between biological and artificial representations (e.g. does the information processing hierarchy in deep nets match the anatomical hierarchy of mouse visual cortex?). Along the way, we catalogue a number of models (including vision transformers, MLP-Mixers, normalization free networks, Taskonomy encoders and self-supervised models) outside the traditional circuit of convolutional object recognition. Taken together, our results provide a reference point for future ventures in the deep neural network modeling of mouse visual cortex, hinting at novel combinations of mapping method, architecture, and task to more fully characterize the computational motifs of visual representation in a species so central to neuroscience, but with a perceptual physiology and ecology markedly different from the ones we study in primates.

---

[*]Correspondence: conwell@g.harvard.edu; Project Website: github.com/ColinConwell/DeepMouseTrap

35th Conference on Neural Information Processing Systems (NeurIPS 2021).

# 1 Introduction

To date, the most successful models of biological visual cortex are object-recognizing deep neural networks applied to the prediction of neural activity in primate visual cortex [1–5]. Corresponding to the biology not only at the level of individual layers, but across the feature hierarchy, these models are so powerful they can now effectively be used as neural controllers, synthesizing stimuli that drive neural activity far beyond the range evoked by any handmade experimental stimulus [6]. The correspondence of these same models to mouse visual cortex, on the other hand, has proven a bit more tenuous [7, 8], with a recent finding even suggesting that randomly initialized networks are as predictive of rodent visual cortical activity as trained ones [9].

Often implicit in interpretations of these results is the notion that the visual milieu and machinery of mice is simply *different* – something characterized more, perhaps, by brute force predator avoidance and the 'flexible random associations' thought to define senses like olfaction [10] than by the sophisticated active sampling and representational compositionality enabled by primate central vision. And yet, mice do recognize objects [11, 12] – and do engage in other sophisticated visual behaviors [13] that suggest they must have visual solutions that at least functionally approximate the kinds of solutions learned by modern computer vision algorithms. If these models perform well in monkeys, but not in mice, are we overfitting to an artifact? Are the object recognition capabilities of mice simply the byproduct of a representational competence learned through other (even more behaviorally relevant) tasks? Have mice perhaps converged on solutions to visual problems that fundamentally differ from the solutions that undergird the emergent similarity between monkeys and machines? To even begin to answer these questions, we need substantially more comprehensive modeling statistics than we currently have. Our main goal in this work was to provide exactly that – to re-examine at large scale the state of neural network modeling in the visual cortices of mice, using many thousands of neurons, over 110 distinct neural network models, and two methods of mapping models to brain.

We summarize the statistics from our benchmarking survey in five main results:

1. Training matters. The randomly initialized variants of some convolutional architectures fare well when predicting individual neural responses, but representational similarity is always better captured by features learned in service of some task. (Segmentation seems best.)
2. Features of intermediate complexity dominate in the prediction of all cortical sites, but both our mapping methods do demonstrate an upwards gradient in complexity from primary visual cortex onwards that roughly matches the information processing hierarchy proposed elsewhere in the rodent neurophysiology literature.
3. Taskonomic tools that have previously been shown to approximate functional organization in primates fail to strongly differentiate anatomical regions in mice, with the same kinds of tasks dominant across multiple, distinct neural sites.
4. When aggregated in similar ways, representational similarity and neural regression methods capture similar trends in the kinds of feature spaces that best predict the biology.
5. While still far from the overall noise ceiling for this highly reliable neural data, a variety of the artificial deep net models in our survey make predictions only slightly less accurate than 'biological conspecific models' composed of the neurons from other mice.

# 2 Methods

## 2.1 Neural Dataset

For neural data, we use the Allen Brain Observatory Visual Coding[1] dataset [14] collected with two-photon calcium-imaging from the visual cortex of 256 awake adult transgenic mice and consisting of approximately 59,610 unique, individual neurons. Calcium-imaging fluorescence patterns are preprocessed and deconvolved by the Allen Institute[2]. The neurons sampled include neurons from 6 visual cortical areas at 4 cortical depths across 12 genetic cre lines. The visual experiments recorded activity for both artificial images (e.g., diffraction gratings) and 118 natural scenes. We analyze only

---

[1]Available with a non-commercial license under the Allen Institute terms of use:
http://www.alleninstitute.org/legal/terms-use/
[2]More details are available in the whitepapers released with the observatory data:
http://observatory.brain-map.org/visualcoding/transgenic

the latter to ensure comparable inputs to what is typically used in the training of deep nets. Each natural scene is displayed 50 times over the course of an assay.

To ensure an optimal signal to noise ratio, we perform a significant amount of subsetting on the full neural population, beginning by subsetting only excitatory neurons. Recent analyses suggest neural activity throughout mouse visual cortex is often impacted by extraneous, external body movements [15]. For this reason, we subsequently filter out any neurons whose peak responses to the presentation of natural scene images are significantly modulated by the mouse's running speed, using an ANOVA metric provided by the Allen Institute. We further subselect neurons by assessing their split-half reliability across trials (with each split-half constituting 25 of 50 presentations for each image), keeping only those neurons exhibiting 0.8 reliability and above. This thresholding still leaves 6619 neurons for analysis, is in line with prior work on primates, and supports, for example, the construction of cortical representational dissimilarity matrices (RDMs) with split-half reliabilities as high as 0.93. (More details on the relationship between our metrics and neural reliability, including visualizations of some of our results across many degrees of thresholding, can be found in A.4 of the Appendix.)

## 2.2 Model Zoology

To explore the influence of model architecture on predictive performance, we use 26 model architectures from the Torchvision (PyTorch) model zoo [16] and 65 model architectures from the Timm [17] model zoo [18–52]. These models include convolutional networks, vision transformers, normalization-free networks and MLP-Mixer models. For each of these models, we extract the features from one trained and one randomly initialized variant (using whatever initialization scheme the model authors deemed best) so as to better disentangle what training on object recognition affords us in terms of predictive power.

## 2.3 Neural Taskonomy

Model zoology provides decent perspective on the computations related to object recognition, but the responsibilities of the visual cortex (no matter the species) extend far beyond identifying the category of an object. To probe a wider range of tasks, we turn to Taskonomy: a single architecture trained on 24 different common computer vision tasks [53], ranging from autoencoding to edge detection. The model weights we use are from updated PyTorch implementations of the original Tensorflow models [54]. Key to the engineering of Taskonomy is the use of an encoder-decoder design in which only the construction of the decoder varies across tasks. While recent analyses using a similar approach in human visual cortex with fMRI data [55] have tended to focus only on the latent space of each task's encoder, we extract representations *across all layers*, better situating Taskonomy within the same empirical paradigm that has so far defined the modeling of object recognition in the primate brain. For further clarity, we cluster the 24 tasks according to their 'Taskonomic' category — a total of 5 clusters (2D, 3D, semantic, geometric or other) that we further collapse into 4 clusters (lumping the only member of the 'other' category — a denoising autoencoder — in with its closest cousin — a vanilla autoencoder in the '2D' category). These purely data-driven clusters are derived from estimates of how effectively a set of features learned for one task transfer to (or boost the performance in) another task [53]. Use of the Taskonomy models provides a unique opportunity to test variance in training regimes without the confound of simultaneous changes in architecture.

## 2.4 Self-Supervised Models

Full category supervision, while robust in its ability to build representations that transfer well to a variety of tasks, suffers in its neuroscientific relevance as an ethologically plausible mode of learning. Recently, self-supervised models have begun to provide viable alternatives to the representations learned by category-supervised models in both computer vision [56, 57] and neural mapping [58, 59]. Here, we assess 22 self-supervision models from the VISSL model zoo [60], ranging from earlier iterations (e.g. DeepCluster [61]) to modern contrastive learning algorithms (e.g. BarlowTwins and Dino [62–65]). We use these models to assess whether category-supervision, however powerful it is in predicting neural activity, might eventually be supplanted by these more realistic alternatives. 14 of these models have as their base architecture a standard ResNet50; 8 are built atop vision transformers.

## 2.5 Comparing Representations across Biological & Artificial Networks

Two methods predominate in the comparison of neural recordings to deep neural networks: at the most abstract level, one of these compares representational geometries computed across the activations

of many individual neurons [66, 67]; the other attempts to predict the activity of individual neurons directly [67, 68]. Both of these techniques are grounded in the use of image-computable models and a shared stimulus set, but differ in the types of transformation applied to the neural activity generated by those stimuli. Given the difference in both target (neural populations versus individual neurons) and transforms (correlation matrices versus dimensionality reduction) we attempt a variant of each type of analysis here, comparing the two directly on the exact same neural data, with the same models and the same stimulus set, and in a granular, layer-by-layer fashion. (A more comprehensive review of neural mapping methods is provided in Section A.2 of the Appendix.)

### 2.5.1 Representational Similarity Analysis

To compare the representational geometries of a given model to the representational geometries of the brain, we begin by computing classic representational dissimilarity matrices (RDMs) [69]. We compute these RDMS by calculating the pairwise correlation coefficients between the neural response vectors for each image (one for each of the 6 cortical areas surveyed). We then repeat this procedure for the artificial networks, aggregating the responses of the artificial neurons in a given layer, before aggregating them once more into a correlation matrix. We then measure the relationship between the RDMs computed from the biological and artificial networks with a second-order Pearson correlation between the flattened upper triangles of each. The resultant coefficient constitutes the score for how well a given model layer predicts the representational similarity of a given cortical area.

### 2.5.2 Neural Regression (Encoding Models)

To more directly compare the biological and artificial neural activations in our data, we use a style of regression made popular in the modeling of primate visual cortex, epitomized by BrainScore [4]. Variants of this approach abound, but most consist of extracting model activations, performing dimensionality reduction, and then some form of cross-validated penalized or principal components regression. The dimensionality-reduced feature spaces of the model are used as the regressors of the activation patterns in a given neuron. After testing a number of these variants, we settled on sparse random projection for dimensionality reduction (which proved far more computationally efficient than standard PCA, without sacrifice in terms of regression scores), followed by ridge regression (in place of the more frequently used partial least squares regression).

The details of our method (programmed with [70]) are as follows: Given a network, we first extract a predetermined number of sparse random projections (4096, in this case) from the features of each layer — in line with the Johnson-Lindenstrauss lemma for the number of observations (images shown to the mice) in our data set [3]. After extracting these projections, we regress them on the activity of each individual neuron using ridge regression (with a default lambda penalty of 1.0). The use of a penalized regression in this case allows us to monopolize generalized cross-validation (a linear algebraic form of leave-one-out cross-validation), yielding a set of predictions for the activity of each neuron for each image[4]. We then compute the Pearson correlation between the predicted and actual activity for each neuron to obtain a score per neuron per model layer, which we then aggregate by taking the mean of scores per neuron across cortical area.

We verify the efficacy of this method on the publicly available benchmarks of primate BrainScore, where (relative to BrainScore's in-house regression method) we demonstrate provisional gains not only in terms of predictive score (sometimes up to $r = 34\%$), but also in terms of speed and computational efficiency. (Details may be found in Section A.1 of the Appendix.)

### 2.6 Model Rankings

To rank the models according to how well they predict the variance in a given cortical area, we take the max across layers. In effect, this requires that a model 'commit' only one layer to the prediction of each area. In the case of our neural regression metric we call these scores the 'SRP-Ridge Max'; in the case of our representational similarity metric we call these scores the 'RSA Max'. A final mean taken over the SRP-Ridge Max and RSA Max scores per model per cortical area yields our overall model rankings, which serve as the basis for the bulk of our analyses.

---

[3]Note that in cases where the dimensionality of features is less than the number of projections suggested by the lemma, sparse random projections will actually upsample the feature space, rather than downsample it.

[4]The use of generalized cross-validation is particularly beneficial in datasets with fewer probe images, where $k$-fold cross-validation means losing a significant degree of information in each fit.

## 2.7 Non-Neural Network Baselines

Prior to the ascendancy of neural network models, a significant amount of time and craft was invested in the hand-engineering of features to simultaneously facilitate image recognition and capture meaningful subsets of neural variance. In this work, we test how well a small subset of those features are able to explain the variance in rodent visual cortex, using both our neural encoding and representational similarity metrics. Our non-neural network baselines consist of random fourier features [71] (computed specifically to match the dimensionality of our neural network predictors), handcrafted gabor filters and GIST (spatial envelope) descriptors [72].

## 3 Results

### 3.1 How do trained models compare to randomly initialized models?

Previous work in the deep neural network modeling of mouse visual cortex found that a randomly initialized VGG16 predicted neural responses as well as, if not slightly better than, a VGG16 trained on ImageNet [9], suggesting that the neural predictivity of the features produced by a trained object recognition model are perhaps no better than the features produced by a randomly initialized one. Our results, on the other hand, suggest that the neural predictivity of trained versus randomly initialized models more generally depends on both the particular model being tested and the particular method used to produce the mappings between model and brain.

At the level of individual neurons (neural regression), 17 of the 91 model architectures we tested had randomly initialized variants that either matched or outperformed their ImageNet-trained counterparts. Replicating previous findings, we found these 17 architectures to include VGG16, as well as all 3 other VGG variants (11, 13 & 19), AlexNet, the DenseNet architectures (121, 169, 201), and almost all of the normalization-free architectures. Despite this, a paired t-test of the difference in scores across *all* models demonstrates that ImageNet-trained architectures are still overall more performant than their randomly initialized counterparts (Student's $t = 7.74, p = 1.37\mathrm{e} - 11$, Hedge's $\widehat{g} = 0.81$). At the level of emergent representational similarity (RSA), ImageNet-trained models categorically outperform their randomly initialized counterparts, and by a large margin (Student's $t = 22.66, p = 5.81\mathrm{e} - 39$, Hedge's $\widehat{g} = 2.36$).

Taken together, these results strongly affirm that *training matters*, and that randomly initialized features can only go so far in the prediction of meaningful neural variance. Differences between ImageNet-trained and randomly initialized models are shown in Figure 1.

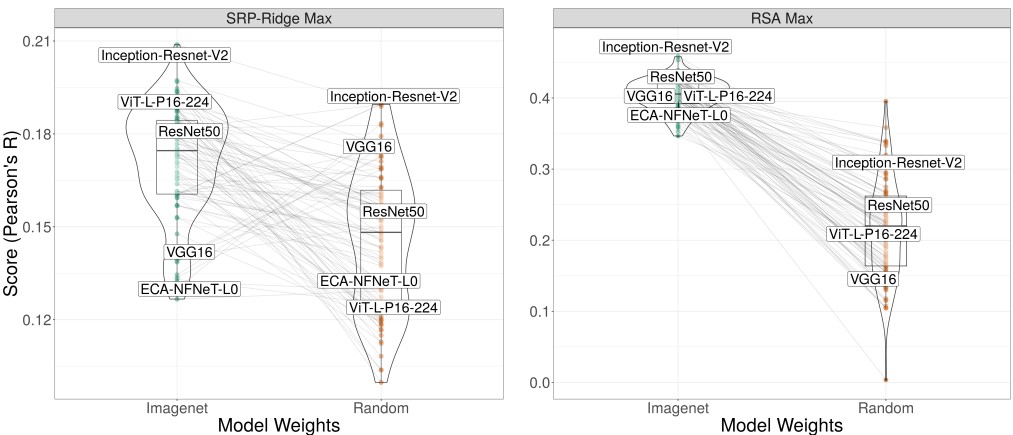

Figure 1: Each pair of connected points in these plots is the score of a trained versus randomly-initialized version of the same model architecture. Here, we've labeled 5 of these pairs for demonstration's sake. The key takeaway of these plots is that ImageNet-trained models significantly outperform their randomly initialized counterparts – majoritively in the case of the SRP-Ridge Max metric, and categorically in the case of the RSA Max metric.

### 3.2 What kinds of architectures best predict rodent visual cortex?

The overall best architecture for predicting mouse visual cortex across both individual neurons (SRP-Ridge) and population-level representation (RSA) was an Inception-ResNet hybrid (Inception-ResNet-V2). There is a small, positive correlation between the depth of a model (the number of distinct layers) for both the RSA-Max metric and SRP-Ridge metric (Spearman's $r = 0.22, p = 0.001$ and $r = 0.192, p = 0.007$, respectively), and a small, negative correlation for the total number of trainable parameters in the RSA Max metric (Spearman's $r = -0.18, p = 0.007$). The latter of these is most likely driven by the relatively poor performance of parameter-dense architectures like VGG.

Markedly, trends previously noted in macaques [73] fail to materialize here. In particular, models with higher top-1 accuracies on ImageNet do not perform significantly better than models with lower top-1 accuracies. This relative parity is driven in large part it seems by newer models like EfficientNets, which across the board have dominant scores on ImageNet, but sometimes middling or poor scores in the predictions of rodent visual cortex we've tabulated here.

Compared to all other architectures, transformers on average fare slightly worse in the RSA Max metric (Student's $t = -3.96, p = 0.004$, Hedge's $\widehat{g} = -0.913$), but moderately better in the SRP-Ridge Max metric (Student's $t = 2.45, p = 0.023$, Hedge's $\widehat{g} = 0.633$).

Strikingly, transformers and MLP-Mixers boast the largest differences between ImageNet-trained and randomly initialized variants in the SRP-Ridge Max metric, with all pairwise t-tests significant at $alpha = 0.05$ after Bonferroni correction for multiple comparisons. This strongly suggests that the advantage of those randomly initialized variants that matched or outperformed their ImageNet-trained counterparts is an advantage conferred by properties of convolutional architectures (e.g., translation invariance), and not necessarily an advantage shared across random feature spaces writ large. (The rankings of these and other architectures may be found in Figure 8 in the Appendix).

### 3.3 What kinds of tasks best predict rodent visual cortex?

The overall best Taskonomy encoder across both the RSA and SRP-Ridge Max is 2D segmentation (ranking second and first respectively; see Figure 9 in the Appendix). At the level of individual neurons (SRP-Ridge), 2D tasks (keypoints, autoencoding, inpainting) dominate. At the level of representational similarity (RSA), all 2D tasks but 2D segmentation fall to the bottom of the rankings, and Semantic tasks (object recognition and semantic segmentation) rise to 2nd and 3rd place.

This reshifting in rank presents a curious case for interpretation, suggesting most likely that while the representations of individual neurons may be coordinated more by the lower level, less abstract features necessary for performing well on most 2D tasks, the overall neural population codes are coordinated more by the parsing of the visual input into ethologically and spatially relevant units via the segmentation and classification tasks. Notably, the original research from which these PyTorch models were adopted offers an auxiliary data point that may anchor this interpretation more concretely. The top 3 models in our RSA Max metric (2D segmentation, object classification, semantic segmentation) are likewise in the top 5 of a ranking the original researchers produced by pitting the Taskonomy encoders against one another as pretrained 'perceptual systems' for reinforcement learning agents learning to navigate a virtual environment (see [54], figure 13 in the appendix). This raises the possibility that the reason these models are optimally predicting the visual neural population code for mice is simply because that code is coordinated in service of navigation.

### 3.4 How do category-supervised models compare to self-supervised models?

Whether with ResNet50 as their base, or a vision transformer, self-supervised models seem to be verging closer and closer to the predictive power of their category-supervised counterparts. Our most predictive self-supervised ResNet50, for example, (a MocoV2 model) effectively matches its category-supervised counterpart in the SRP-Ridge Max metric (with scores of .182 and .184, respectively), while slightly outperforming its category-supervised counterpart in the RSA Max metric (with scores of .422 and .415, respectively). While this single comparison by no means denotes a statistically significant superiority of self-supervised models (which would require training multiple iterations of each), it does begin to provide preliminary evidence for parity.

### 3.5 How well do non-neural network baselines predict rodent visual cortex?

Non-neural network baselines somewhat uniformly fail to predict neural activity as accurately as deep net features (though see Section A.10 of the Appendix for a counterexample). We tested three baselines: 1) a bank of Gabor filters of applied to 8x8 grids of each image; 2) the PCs of the resultant feature matrices (i.e. the Gist descriptors [72]); 3) and the max across 600 iterations of 4096 random Fourier features (a dimensionality matching that of our SRPs). Ridge regressed with generalized cross-validation, these feature models yield average scores of 0.07, 0.06 and -0.014, respectively. Compared via representational similarity, they yield average scores of 0.20, 0.25 and 0.011.

### 3.6 How 'deep' are the layers that best predict rodent visual cortex?

Echoing previous results [7, 8], we find across all ImageNet-trained architectures, regardless of metric, that the features most predictive of rodent visual cortex are found about a third of the way into the model (though see Section A.5 of the Appendix for some caveats). These early to intermediate visual features go beyond basic edge detection but are far from the highly abstracted representations adjacent to final fully connected layers. Across Taxonomy encoders, 2D & Geometric tasks yield their best features in earlier layers; 3D & Semantic tasks yield their best features in more intermediate and later layers. Note that these aggregate motifs do not preclude subtler differences across cortical area, which we discuss in the section below.

### 3.7 Are there differences in model predictions across cortical area?

In this work, we address this question from two perspectives: that of hierarchy and that of function.

In primate visual cortex, it is common consensus that there exists a distinct information processing hierarchy along the ventral visual stream [74–76], with posterior sites like V1 and V3 defined by features like oriented edge detectors, and more anterior sites like V4 and IT defined by more complex morphologies. While there continues to be some debate as to whether a similar hierarchy exists in rodent visual cortex, a large body of anatomical, functional and physiological work [77–83] has coalesced around a meaningful hierarchy that consists first of a ventral / dorsal split after primary visual cortex (VISp), with VISp leading to VISl in the ventral stream and VISp leading to VISrl - VISal - VISpm - VISam in the dorsal stream. Strikingly, our modeling does seem to provide corresponding evidence for this circuit in the form of a data-driven hierarchy produced purely by taking the median depths of the model layers that best predict the neural activity in each of these cortical areas, and assessing for difference across them. A nonparametric ANOVA shows an overall difference in depth across cortical area to be significant for both our SRP-Ridge metric (Friedman's $\chi^2 = 34.08$, $p = 2.29e - 06$, Kendall's $\widehat{W} = 0.04$) and our RSA metric (Friedman's $\chi^2 = 37.05$, $5.86e - 076$, Kendall's $\widehat{W} = 0.06$). Subsequent pairwise comparisons show many of the differences that underlie this group-level effect to be differences between earlier and later layers of the information processing hierarchy established in the literature. (For further details, see Figure 2.)

Other differences across cortical area that we might expect are differences driven by function. Research into primate visual cortex over the last two decades has unveiled a significant degree of functional organization over and above purely anatomical organization [84–86], with distinct subregions defined in large part by their differential activity in response to different kinds of stimuli. To try and replicate this in mouse visual cortex we search for Taxonomic organization, a proxy of functional organization wherein distinct neural sites are better or worse predicted by the features from different taxonomy encoders. Curiously, and in contrast to previous findings in human fMRI [55], it seems to be the case that the scores of different Taxonomic clusters are relatively consistent across cortical area; see Figure 3.) This suggests that mouse visual cortex *may* be more functionally (or Taxonomically) homogenous than primate visual cortex, with anatomical descriptors providing little to no cue of functional difference – though this seems unlikely given other analyses we've performed showing greater similarities of neurons *within* cortical site than *between* cortical site (see Section A.11 for details). Another (more likely) alternative is that the tasks of computer vision are just not so neatly mapped onto the tasks of biological vision in mice.

### 3.8 How do the predictions compare across RSA and neural regression?

While prior work has addressed this question theoretically [87], it's rarely the case that representational similarity and neural regression are compared directly and empirically. Here, we compare our RSA and SRP-Ridge metric both at the level of overall rankings (taking the max across layers) and at

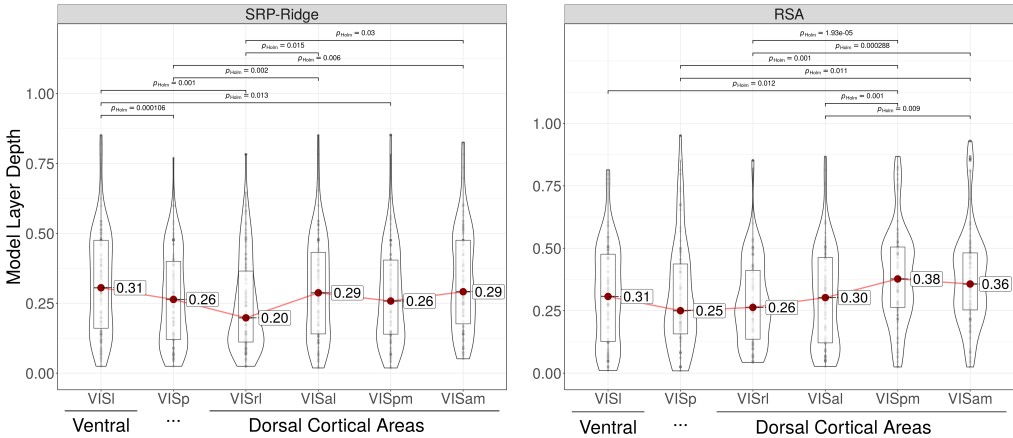

Figure 2: In these plots, we demonstrate a purely data-driven approach for recapitulating the proposed information processing hierarchy in mouse visual cortex. On the x axis in these plots are the 6 cortical areas in our optical physiology sample, arranged roughly according to the schematic proposed by [79] (Figure 3C). On the y axis is the average depth from 0 (the first layer) to 1 (the last layer) of the layer in each model that best predicts a given cortical area. Each point is an individual model. The horizontal brackets are drawn only in the case of a significant difference in a Games-Howell (Holm-corrected) pairwise comparison test. While noisy, these plots demonstrate that the deeper into the information processing hierarchy a given cortical area is, the deeper the average depth of the maximally correspondent model layers (and more complex the features). Even in those cases where we see a violation of the order suggested by the physiology (as in the case of the difference between in median depths between VISp and VISrl in the SRP-Ridge metric), these differences are insignificant. All *significant* differences point in the direction proposed by the physiology.
.

the level of individual layers, the latter of which provides a much more detailed assessment of how different feature spaces map to cortical representation.

In terms of overall rankings, the Spearman rank order correlation between the two methods is either 0.56 ($p = 8.36 \times 10^{-19}$) or 0.59 ($p = 3.17 \times 10^{-12}$), depending on whether you include or exclude the randomly initialized architectures. In terms of layer by layer comparisons, we decompose the Spearman correlation across distinct combinations of model and cortical area. The average coefficient between the two methods, along with bootstrapped 95% confidence intervals is 0.468 [0.447,0.489] or 0.626 [0.613,0.639], again depending on the inclusion or exclusion of the random models. This suggests a significant degree of overlap between the kinds of features that optimally predict the representations of both individual neurons and neural populations. Of course, the averages here do obscure some meaningful subtrends and idiosyncrasies. For details, see Figure 4.

### 3.9 How well are we doing overall in predicting mouse visual cortex?

The overall best model in any cortical area across either of our metrics is unsupervised 2D segmentation in anterolateral visual area (VISal), with an RSA Max score of 0.538. The (Spearman-Brown) splithalf reliability of the RDM for this area (an effective proxy of its explainable variance) is 0.89. This means our most predictive model in any cortical area across any metric is little more than halfway to the noise ceiling.

Of course, it's possible this noise ceiling is a bit too strict. Instead of requiring the model to predict the neural data as well as the neural data predicts itself, another possible target to which we might recalibrate is the relative performance we would expect if (instead of an artificial neural network) we used the responses of another biological network as the model to predict neural activity. Inspired by recent work [88], and to better contextualize the scores of our SRP-Ridge metric, we attempted a version of this here. To compute this reference, we proceeded again neuron by neuron using the exact same neural regression method (dimesnonality reduction, and hyperparameters) described in Section 2.5.2, but instead of using the responses of a deep net layer as the predictors in our ridge regression, we used the responses of the neurons from the same cortical area in all other mice (conspecifics) across the donor sample. Conceptually, this 'intermouse score' represents how well we might do if our model of a given mouse brain were other mouse brains.

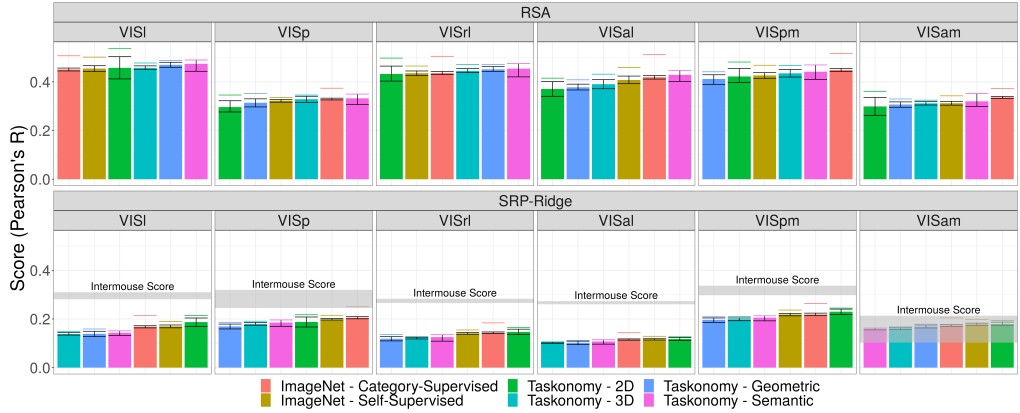

Figure 3: A mosaic of scores across distinct kinds of models in different cortical areas. Each bar is the average score across all models of a certain kind of training regimen or task affinity cluster (denoted by color). Error bars are 95% bootstrapped confidence intervals across models. The dashed lines above each bar are the scores of the best performing models in that category. Within each facet, the averages are arranged in ascending order, with the best performing models on the right and the worst performing on the left. For the SRP-Ridge metric, we've included a reference point we call the 'intermouse score', which leverages the neural activity of other mice to predict the activity in the cells of a target mouse. The band corresponding to the intermouse score is the 95% confidence interval across the scores for all cells in a given cortical area. Intermouse scores (as a sort of noise ceiling) tend to be a much more attainable target than the average splithalf reliability, which ranges from 0.886 to 0.906 across cortical area. There are a few takeaways from this plot: first, we see that there do not seem to be major 'taskonomic' dissociations across cortical area. 2D tasks (driven by unsupervised segmentation) dominate in the SRP-Ridge max metric, and despite a similarly strong showing from unsupervised segmentation, Semantic tasks are the evident superlative in the RSA Max metric. Second, we see that our models are in some cases on the verge of providing estimates of neural activity as well suited to the prediction of a given biological neuron as the reweighted activity of dozens of other biological neurons sampled from the same species. Finally, we see that (overall) there tend to be very few divergences or punctuated gaps in the average scores across model – underscoring that even sometimes *substantive* differences in the tools of computer vision do not always translate immediately to differences in neural predictivity.

Averaging across both cortical area and model, the average distance (with 95% bootstrapped confidence intervals) between the best performing deep net feature spaces and the mean of the intermouse scores (expressed in the same units of Pearson's $r$ we've used heretofore) is 0.0985 [0.0940, 0.103]. Compare this to the same distance computed relative to the splithalf reliability: 728 [0.726, 0.731]. On average, then, while our artificial models are capturing only a fraction of the total explainable variance relative to the splithalf noise ceiling, they're verging increasingly close to the predictive threshold suggested by the reweighting of biological neurons from the same species. The performance of models relative to the intermouse score may be seen in the lower half of Figure 3.

## 4 Discussion

Our intent with this work was to provide a preliminary atlas for future ventures into the deep neural network modeling of rodent visual cortex. To this end, we have deliberately invested in introspective analyses of the tools we used, as well as the curation of deep neural networks we hope will provide informative waypoints. Obviously, the atlas is far from complete. Other model classes like recurrent models [89, 90], equivariant models [91], and robotic models (e.g. for visual odometry [92]) are promising candidates for inclusion in future benchmarks, and our neural encoding & representational similarity metrics are just two of many variants.

Nevertheless, the results we have presented here indicate that neural recordings from the visual brains of mice can (with care and caution) be compared to deep neural networks using many of the same tools we've used to better characterize the visual brains of monkeys. Having as reference two animal models that occupy very different ecological niches and are separated by tens of million

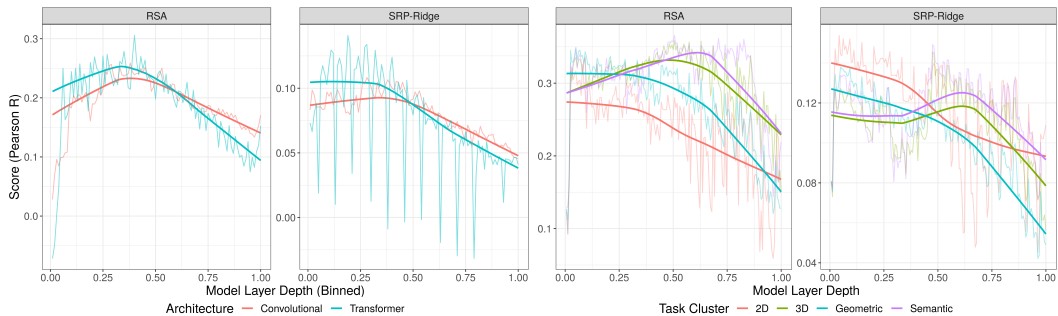

Figure 4: Aggregate layer by layer comparisons across ImageNet-trained models (left) and Taskonomy models (right). The horizontal axis shows relative model layer depth, from the first to the last layer. Because the models on the left vary significantly in size, we discretize the relative depths into bins of size 0.01 (100 bins). The jagged, semitransparent lines are the mean scores at a given depth. (These are particularly jagged for transformers due to the heterogeneous nature of the computational blocks, which engenders large troughs in predictive power). The opaque, smoothed trend lines are the outputs of locally weighted linear regressions (lowess) computed at each point with a span of 2/3 the width of the axis. Plots show that convolutional networks and transformers have similar predictive power at similar depths despite their significant computational differences. (right) Features most predictive of cortical responses across the Taskonomy models vary in their relative depth. The reversal in the ranks of 2D models between RSA Max and SRP Ridge Max is seen as a vertical shift in the intercept of the curve relating depth to score – though note that the slope and shape of the curve remain similar across metrics. 2D models are most predictive in the early layers, but are nevertheless superseded by other tasks in the later layers at the level of representational similarity.

years of evolution makes it far more likely that insights into vision gleaned across both are actually fundamental to perceptual meaning-making and not just some idiosyncratic quirk specific to any one evolutionary trajectory. Primate and rodent vision do differ rather drastically, even in fairly basic ways: mice lack a fovea, have a retina dominated by rods for vision under low light, and spatial acuity less than 20/1000 [93], making their primary visual system more akin to the primate peripheral system – and making it all the more curious that the same models explain decent amounts of variance in both. The differences between the species, it seems, may not be so irreconcilable at the level of modeling, but only with future work more carefully controlling for distinct aspects of each organism's unique physiology (see Section A.5) can more concrete conclusions of this kind be made.

Beyond considerations of distinctive physiology is the indispensable point that perceptual systems should always be considered in service of behavior. It's possible that mice mostly rely on vision as a sort of broad bandpass filter for lower-frequency, dynamic stimuli the animal can then flee, fight, or further investigate with its whiskers — perhaps its most sophisticated sensory organ [94]. Another possibility is that mice use vision to facilitate navigation. The dominance in our Taskonomy results of 2D segmentation, object recognition and semantic segmentation (all tasks that have elsewhere been shown to provide effective, transferable features for the simulation of robotic navigation) provide some evidence for this. Of course, the behavioral roles of rodent vision may very well be manifold. Understanding this plurality in a readily available model species could in the end be key for bridging the gaps that remain between biological and computer vision [95]. The unparalleled access, resolution, and control afforded by rodent neuroimaging have already revolutionized our understanding of the relationship between perceptual representation and behavioral output. Combined with novel methods like the embedding of neural networks in virtual agents [96] in ecologically realistic environments, this kind of data may well provide a testbed for better situating the tasks of computer vision in the broader behavioral context of agentic scene understanding.

In summary, only novel combinations of architecture, task and mapping will help to explain the highly reliable neural variance we've yet to explain in our current survey. Already this recombination is under way: Shi et al. [97] have created a custom CNN designed specifically to match (processing stage by processing stage) the anatomy of rodent visual cortex, while Nayebi et al. [88] have combined the power of self-supervised learning with smaller, shallower architectures to more fully account for the ethological realities of rodent behavior and the differences in computational bandwidth that shape and constrain their visual systems. More work of this variety will be necessary to more fully model the rich diversity and fiendish complexity of biological brains at scale – even the very smallest ones.

### 4.1 Acknowledgements

We thank Martin Schrimpf, Tiago Marques, Jim DiCarlo, as well as many others on the BrainScore team for helpful discussion, feedback, and inspiration. We would also like to thank the Allen Institute founder, Paul G. Allen, for his vision, encouragement, and support.

### 4.2 Code Availability

More results and code for the replication of our analysis may be found at this GitHub repository: github.com/ColinConwell/DeepMouseTrap (License GPL v2)

### 4.3 Compute Required

We used a single machine with 8 Nvidia RTX 3090 GPUs, 755gb of RAM, and 96 CPUs. GPUs were used only for extracting model activations, and could (without major slowdown) be removed from the analytic pipeline. Dimensionality reduction and regression computations were CPU and RAM intensive. Replicating all of our results would take approximately two weeks on a similar machine.

### 4.4 Ethics Statement

Lest our science forget the life that powers it, we must note that behind the phenomenal dataset provided by the Allen Institute are 256 laboratory mice, each of which was subjected to multiple surgeries, a highly invasive neuroimaging technique and genetic engineering. The moral parameters of this particular praxis of neuroscience are contentious, and not without reason. While we believe centralized, comprehensive and (most importantly) public datasets like those provided by the Allen Institute may actually decrease the total number of laboratory animals required for similar kinds of empirical projects, we acknowledge with solemnity the cost to life required.

### 4.5 Funding Statement

This work was supported by the Center for Brains, Minds and Machines, NSF STC award 1231216, the MIT CSAIL Systems that Learn Initiative, the CBMM-Siemens Graduate Fellowship, the MIT-IBM Watson AI Lab, the DARPA Artificial Social Intelligence for Successful Teams (ASIST) program, the United States Air Force Research Laboratory and United States Air Force Artificial Intelligence Accelerator under Cooperative Agreement Number FA8750-19-2-1000, and the Office of Naval Research under Award Number N00014-20-1-2589 and Award Number N00014- 20-1-2643. The views and conclusions contained in this document are those of the authors and should not be interpreted as representing the official policies, either expressed or implied, of the U.S. Government. The U.S. Government is authorized to reproduce and distribute reprints for Government purposes notwithstanding any copyright notation herein.

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
