# A   Appendix for Neural Regression, Representational Similarity, Model Zoology & Neural Taskonomy at Scale in Rodent Visual Cortex

## A.1   Does our neural regression method work?

To ensure our neural regression method works, we verify its efficacy on a known benchmark: the activity of 256 cells in the V4 and IT regions of two Rhesus macaque monkeys, a core component of BrainScore [4]. BrainScore's in-house method involves a combination of principal components analysis (for dimensionality reduction) and $k$-fold cross-validated partial least squares regression (for the linear mapping of model to brain activity). Here, we exchange principal components analysis for sparse random projection and partial least squares regression for ridge regression with generalized cross-validation. We compute the scores for each benchmark in the same fashion as BrainScore: as the Pearson correlation coefficient between the actual and predicted (cross-validated) activity of the biological neurons in the V4 and IT samples.

Taking for example a standard AlexNet architecture, our neural regression method yields gains of 16.5% (from 0.550 to 0.641) & 16.9% (from 0.508 to 0.593) on reported scores for V4 and IT, respectively. Across 6 other Torchvision architectures we tested with scores posted on the BrainScore leaderboard, our method yields gains on average of 13% for V4 and 23% for IT, and at its best yields a gain of 34% for SqueezeNet1-0 predicting IT. Though these scores are provisional (since official BrainScore results involve an additional step of validation – 'commitments' – on data not publicly available), we consider this a strong validation of our neural regression method, which is both less computationally expensive, far faster and (to the extent that the generalized cross validation represents an optimal approximation of how well the mappings fit to our models might generalize to novel biological samples) more accurate than the combination of PCA and PLS. (Speed tests may be found in Section A.3 of this Appendix.) Figure 5 shows the scores for the full subset of models we tested on primate BrainScore.

| Model | Weights | V4 | IT |
|---|---|---|---|
| alexnet | ImageNet | 0.6413 | 0.5939 |
| densenet121 | ImageNet | 0.6555 | 0.6298 |
| googlenet | ImageNet | 0.6551 | 0.6278 |
| mnasnet0_5 | ImageNet | 0.6554 | 0.6228 |
| mnasnet1_0 | ImageNet | 0.6632 | 0.6398 |
| mobilenet_v2 | ImageNet | 0.6592 | 0.6295 |
| resnet101 | ImageNet | 0.6690 | 0.6193 |
| resnet18 | ImageNet | 0.6492 | 0.6233 |
| resnet34 | ImageNet | 0.6560 | 0.6282 |
| resnet50 | ImageNet | 0.6649 | 0.6216 |
| resnext50_32x4d | ImageNet | 0.6474 | 0.6285 |
| shufflenet_v2_x0_5 | ImageNet | 0.6530 | 0.6087 |
| shufflenet_v2_x1_0 | ImageNet | 0.6538 | 0.6296 |
| squeezenet1_0 | ImageNet | 0.6652 | 0.6157 |
| squeezenet1_1 | ImageNet | 0.6602 | 0.6061 |
| vgg11 | ImageNet | 0.6735 | 0.6202 |
| vgg13 | ImageNet | 0.6748 | 0.6201 |
| vgg16 | ImageNet | 0.6750 | 0.6260 |
| vgg19 | ImageNet | 0.6733 | 0.6250 |
| wide_resnet50_2 | ImageNet | 0.6606 | 0.6355 |

Figure 5: Scores for a subset of models tested on the macaque V4 & IT benchmarks of BrainScore. Corresponding scores for models in our set that overlap with models tested by the BrainScore team may be found at: https://www.brain-score.org/

## A.2   Deeper Dive: How do our neural benchmarking methods compare to others?

In the main analysis, we roughly group existing methods for comparing the responses of deep neural networks to neural responses recorded from brain tissue into two categories: neural regression and representational similarity analysis. In reality, this division is often not so neatly dichotomous. Some of the first brain-to-network comparisons availed themselves of both these methods simultaneously;

Yamins et al. [1] citing Carandini et al. [68] and Kriegeskorte et al. [66] used linear regression for mapping responses in individual neural sites and representational similarity analysis for populations. Other seminal work comparing deep nets to primate visual cortex pioneered distinctive variants of each. Güçlü and van Gerven [67] employed regression in the form of encoding models to assess the hierarchical correspondence between earlier and later layers of processing in vivo and silico. Khaligh-Razavi and Kriegeskorte [98] built representational dissimilarity matrices by "remixing" and "reweighting" model features according to their performance in a support vector machine classifier trained on major categorical divisions in the stimulus set. Zhuang et al. [58] citing Klindt et al. [99] uses a form of masked regression to better account for spatial information (e.g. properties of the receptive field) in the target feature spaces. In the context specifically of comparisons to rodent neuro-physiology, Cadena et al. [9]'s neural encoding method predicts spike rate with a core feature model (VGG16) in tandem with a "shifter" network and "modular" network that correct for extraneous influences on recorded brain activity (including eye movements and running speed). A possible third strain of methods that doesn't fit so neatly into the binary of regression versus representational similarity are canonical correlation and alignment methods. These techniques leverage what is often assumed to be an underlying latent space of similarity shared across divergent high-dimensional datasets to assess (via projection) the shared variance between them. Canonical correlation and alignment methods are popular in both the machine learning [100, 101] and neuroimaging communities [102], but have so far been applied mostly to comparison within, rather than across, domains and neural substrates. The relative advantages of these various approaches as they pertain to characterizing the representational structure of biological brains is largely uncertain, with a comprehensive comparison of techniques on the same dataset seemingly absent from the literature.

The current standard for high-throughput benchmarking of neural data on neural models is perhaps that of Schrimpf et al. [4] in BrainScore, a method that consists of a partial least squares (PLS) regression fit individually to each neural site (in their case, a cluster of neurons around a given electrode in a microarray), wherein the regressand is the responses from that site and the regressors are the principal components of a target model's feature space. The end product of this process is a Pearson correlation coefficient (unadjusted or reliability-corrected) quantifying the relationship between actual neural activity and the neural activity predicted by the linear mapping. While effective, this combination of principal components analysis and partial least squares regression tends to be a computationally expensive process – often prohibitively so in the absence of cloud or cluster computing. The final approach we use in the primary manuscript is a more computationally efficient version of this process. The reasoning behind the particular neural regression we use (assessing the tradeoff between accuracy and computational traction) may be found in the section below.

### A.3 How do different neural regression methods trade off in terms of speed & accuracy?

Given the many variants of neural regression used in the analysis of the human and nonhuman primate brain (and to a lesser extent the rodent brain), we experimented with a number of possible approaches before settling on the one detailed in the primary manuscript. Attempting to directly mirror the approach described in Schrimpf et al. [4], we began with a method combining principal components analysis for dimensionality reduction with partial least squares regression for neural prediction. So as to capture more dimensions of variance in a given model's feature spaces, and not 'double dip' meaningful dimensions of variance with the regression to follow, Brain-Score computes a set of principal components on the features from an auxiliary set of held-out ImageNet images, then extracts the loadings of the features from the target stimulus set on these same components. These loadings are subsequently made the regressors in a partial least squares regression of 25 components, with a given neural site (the activity from a microelectrode array) as the regressand. The most computationally intensive step of this process is the calculation of the PCA on the features from the auxiliary Imagenet images – requiring in larger models like VGG16 upwards of 450GB of RAM for a single layer. The prohibitively large expense of this PCA prompted us to search both for alternative dimensionality reduction techniques, as well as for the possibility of extracting fewer total dimensions with whatever technique we chose. A summary of the outputs of this search (using only a small subset of models and a fraction of our total neural data) may be found in Figure 6.

What this search made clear (at least in the context of our specific neural data and stimulus set) is that approaches involving PCA were doubly suboptimal, taking orders of magnitude longer to compute, and actually costing a nontrivial portion of score. In the PLS regressions as well, it quickly became clear that 10 components yielded scores comparable (if not equivalent) with those of twice or thrice as many components, suggesting nothing was to be gained from more components apart

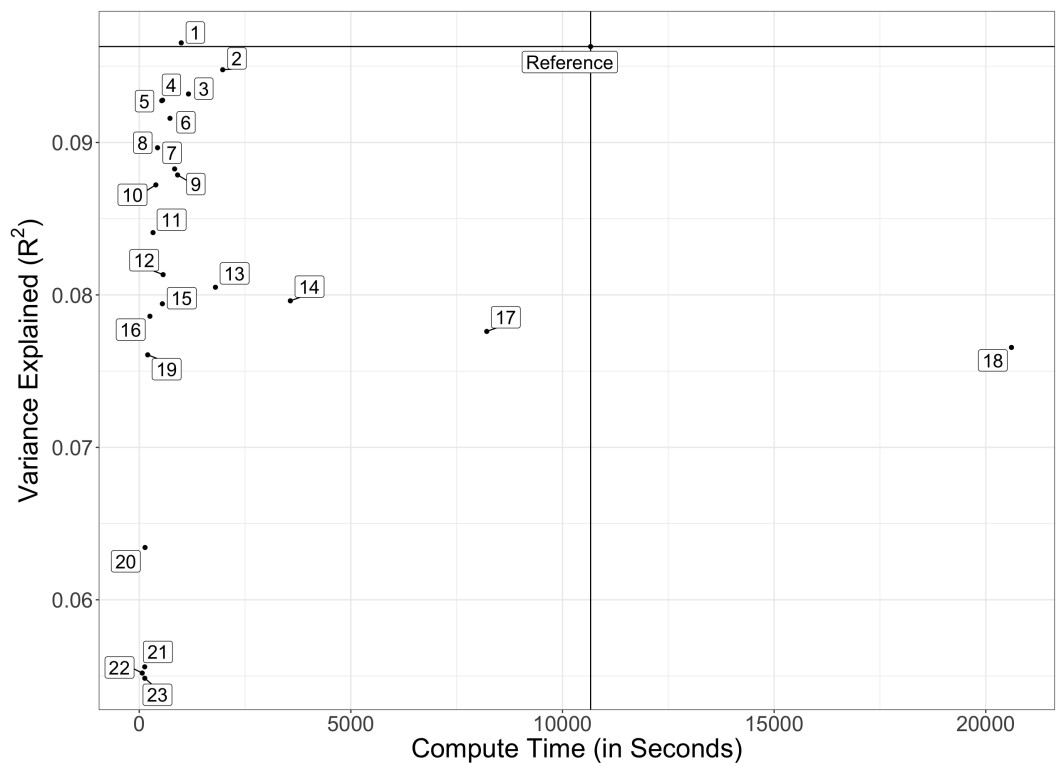

**PLS Regression with 25 Components and Different Dimensionality-Reduced Regressors**

| ID / Rank | Dimensionality Reduction Technique Applied to the Regressors |
|---|---|
| Reference | No Dimensionality Reduction; All Features Included |
| 1 | 1:8192 SRPs calculated directly on the Stimulus Set |
| 2 | 1:8192 Feature SRPs calculated with 9216 ImageNet Images |
| 3 | 1:4096 Feature SRPs calculated with 9216 ImageNet Images |
| 4 | 1:4096 SRPs calculated directly on the Stimulus Set |
| 5 | SRPs corresponding to Embedding Quality of 0.1 |
| 6 | 2048 Randomly Selected Features |
| 7 | 1:2048 Feature SRPs calculated with 9216 ImageNet Images |
| 8 | 4096 Randomly Selected Features |
| 9 | 1:1024 Feature PCs calculated with 1024 ImageNet Images |
| 10 | 1:2048 SRPs calculated directly on the Stimulus Set |
| 11 | 2048 Randomly Selected Features |
| 12 | 1:1024 Feature SRPs calculated with 9216 ImageNet Images |
| 13 | 1:1024 Feature PCs calculated with 9216 ImageNet Images |
| 14 | 1:2048 Feature PCs calculated with 9216 ImageNet Images |
| 15 | 1:1024 Feature SRPs calculated with 1024 ImageNet Images |
| 16 | 1:1024 SRPs calculated directly on the Stimulus Set |
| 17 | 1:4096 Feature PCs calculated with 9216 ImageNet Images |
| 18 | 1:8192 Feature PCs calculated with 9216 ImageNet Images |
| 19 | SRPs corresponding to Embedding Quality of 0.25 |
| 20 | SRPs corresponding to Embedding Quality of 0.5 |
| 21 | SRPs corresponding to Embedding Quality of 0.75 |
| 22 | All PCs calculated directly on the Stimulus Set |
| 23 | SRPs corresponding to Embedding Quality of 0.99 |

Figure 6: Variance explained versus compute time for a PLS Regression with 25 components and different types of dimensionality reduction techniques applied to the regressors. Descriptions of the dimensionality reduction steps associated with each data point are provided in the table above. The vertical and horizontal ablines triangulate a reference point for all methods: that is, a regression in which all features from a given model layer are used simultaneously without dimensionality reduction. Notice, that at least one PCA-based method (18) takes longer than this full regression.

from longer compute times. Finally, while somewhat less definitive than the 2 previous points, our tests did suggest that using 4096 sparse random projections was roughly comparable to using 8192 sparse random projections – and translated to about 0.75 of the compute time per projection. When in an additional test a ridge regression performed in roughly 0.29 seconds what a PLS regression with 10 components performed in roughly 1.31 seconds (representing an over 200% percent gain in speed, and consistent across 10 iterations), while producing only a 0.004 difference in average $R^2$, we abandoned PLS regression entirely in favor of ridge regression. Switching to ridge regression meant we could also make use of generalized cross-validation [70, 103], cutting the time required for $k$-fold cross-validation from roughly 1.05 seconds to 0.25 seconds. While it is most certainly the case that these tests were not comprehensive enough in terms of models or neural data to cover the full range of contingencies and idiosyncrasies of analysis, we felt the empirical justification for the use of a sparse random projection and ridge regression approach was sufficient. In future work, we intend to expand our testing regimen to see if the empirical advantage holds in a wider range of cases.

### A.4  How does reliability thresholding impact our benchmark scores?

In the main analysis, we subselected from the greater pool of available neurons only those neurons with split-half reliabilities of 0.8 and above. In Figure 7, we show the impact of different degrees of thresholding on the scores for a majority of our models.

### A.5  How do we control for the influence of receptive field and visual acuity (resolution)?

The answer, in brief, is that we don't – at least not explicitly. We rely instead on the reweighting procedure inherent to the neural regression method or on averaging across population responses in the representational similarity analysis to account for these properties implicitly. This may or may not be to our detriment. Recent work in both primates and rodents [9, 105–107] suggests that controlling for these factors explicitly (for example, by reshaping, translating or downsampling the resolution of the input) can in certain cases augment the strength and interpretability of the correspondence between the biological and artificial neural substrate. When designing the pipeline for the current study, however, we found that reducing the resolution of the input image by a factor of 2 and a factor of 4 (downsampling from [224,224] to [112,112] and [64,64]) in a representative sample of 12 trained and randomly-initialized models tended only to slightly shift the *absolute* depth of the features that best predict the neural activity, but changed neither the relative scores across models, nor the relative depths of the layers corresponding to different cortical areas. In future work, we hope to revisit this manipulation with a more representative sample of architectures and more vigilant consideration of how these manipulations interface with the input transforms required when extracting the features of pretrained models. Best practice for this interfacing remains unclear in the absence of more comprehensive empirical controls. The use of *custom models* (as in Nayebi et al. [88] and Shi et al. [97]), on the other hand, does not carry with it the same concerns as the use of pretrained models, and seems a promising path forward for probing the effects of input manipulations directly.

### A.6  Does training or architecture matter more for better prediction?

The range in scores between the best and worst performing model architecture trained on ImageNet is 0.209 to 0.121 (0.088) for the SRP-Ridge Max and 0.458 to -0.117 (0.575) for the RSA Max metric (excluding the two normalization-free architectures that produced negative scores and are otherwise significant outliers, the range is more like 0.458 to 0.347 (0.111)); the range between the best and worst performing model in Taskonomy is 0.190 to 0.126 (0.064) for the SRP-Ridge Max and 0.440 to 0.331 (0.109) for RSA Max. These results suggest that neither architecture nor task has a statically meaningful edge in augmenting neural predictivity. Moreover, it's worth noting that the rankings in terms of both task and architecture show only minor differences between the best-performing and the next-best-performing models – with no major increases in performance evident across different design design and training choices, only marginal relative gains.

### A.7  Addendum: What kinds of architectures best predict rodent visual cortex?

Figure 8 provides the rankings for all 'model zoology' architectures in our survey.

### A.8  Addendum: What kinds of architectures best predict rodent visual cortex?

Figure 9 provides the rankings for all taskonomy encoders, clustered by task transfer affinity.

### A.9 Addendum: How do self-supervised models compare to category-supervised models?

Figure 10 provides the rankings for all the self-supervised models in our survey, along with reference points for architecture-matched category-supervised models.

### A.10 Addendum: How well do non-neural network baselines predict rodent visual cortex?

The averages we report in the main analysis with respect to two of our non-neural network baselines (gabors and GISTPCs) do obscure differences across cortical area (and cortical layer). One of the most conspicuous examples is in the case of the scores for our bank of gabors in layer 6 of primary visual cortex (VISp). At $r = 0.165$, (according to the SRP-Ridge) metric, the predictions produced by these features are competitive with the those of both trained and random deep net models.

In future work, we intend to expand our set of non-neural network baselines, incorporating perhaps the larger, more robust set of handcrafted Gabors used in custom CNN models like VOneNet [108].

### A.11 Can we visualize functional specialization even if we can't characterize it?

In the main analysis, we show that (in general) taskonomy models fail to differentiate one cortical area from another in terms of predictivity. Here, we demonstrate this is not necessarily because these areas aren't functionally specialized. Comparing neurons *within* a cortical site to neurons *between* using both a pure correlation-based measure, and the same style of neural regression we use in the main analysis, we show that neurons from the same site are categorically better predictors than neurons from other sites, connoting a stronger representational correspondence to neurons in anatomical proximity. Results from this analysis are available in Figure 11.

### A.12 Are there differences in model predictions across *genetic cre line*?

In the main analysis, we aggregate neurons by anatomical region (cortical area); another method of aggregating neurons is by genetic cre line. Aggregating in this way changes the overall focus of the benchmarking, from asking 'where' certain models fare best in predicting visual cortical activity to 'with what cell types'. The kinds of representational idiosyncrasies that characterize different cell types are beyond the scope of this paper. Nevertheless, as a sampler for those interested, aggregate Taskonomy scores across cre line are provided in Figure 12.

### A.13 Glossary of Visual Cortical Areas in Mouse Brain

Reproduced in Figure 13 is a glossary of visual cortical areas in the mouse brain. More information about the Allen Brain Observatory visual coding dataset may be found at their website: http://observatory.brain-map.org/visualcoding

### A.14 Taskonomy Task Definitions

Reproduced in Figure 14 are Taskonomy's official definitions of its constituent tasks. Further information is available at their website: http://taskonomy.stanford.edu

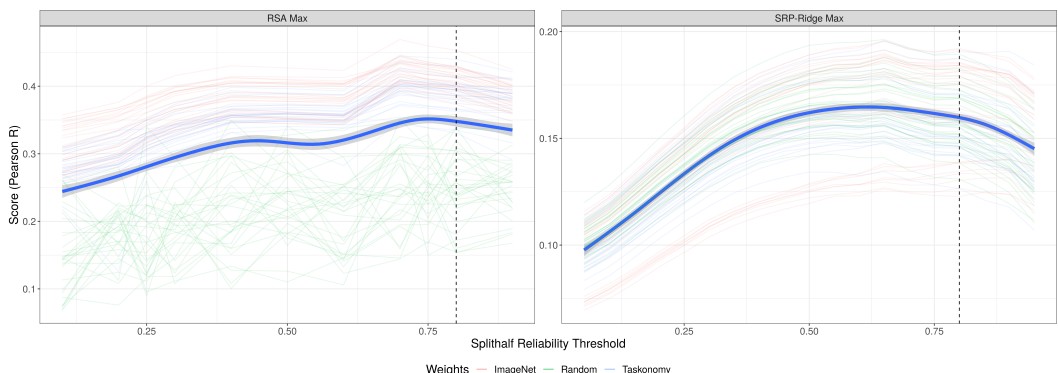

Figure 7: Scores for a variety of models with both the RSA Max (left) and SRP-Ridge Max (right) metrics at different levels of reliability thresholding. The jagged, semitransparent lines are the scores for individual models. The smooth, opaque line is the output of a generalized additive smoother fit across all models. (Error bars are bootstrapped 95% confidence interval across models). The dotted vertical line is the threshold we use in the main analysis. Based on these results, one might argue a more performative threshold would have been closer to 0.7 or 0.75. In the future, we plan to more closely emulate methods designed to derive the optimal threshold empirically (e.g. reliability-based voxel selection in human fMRI [104]).

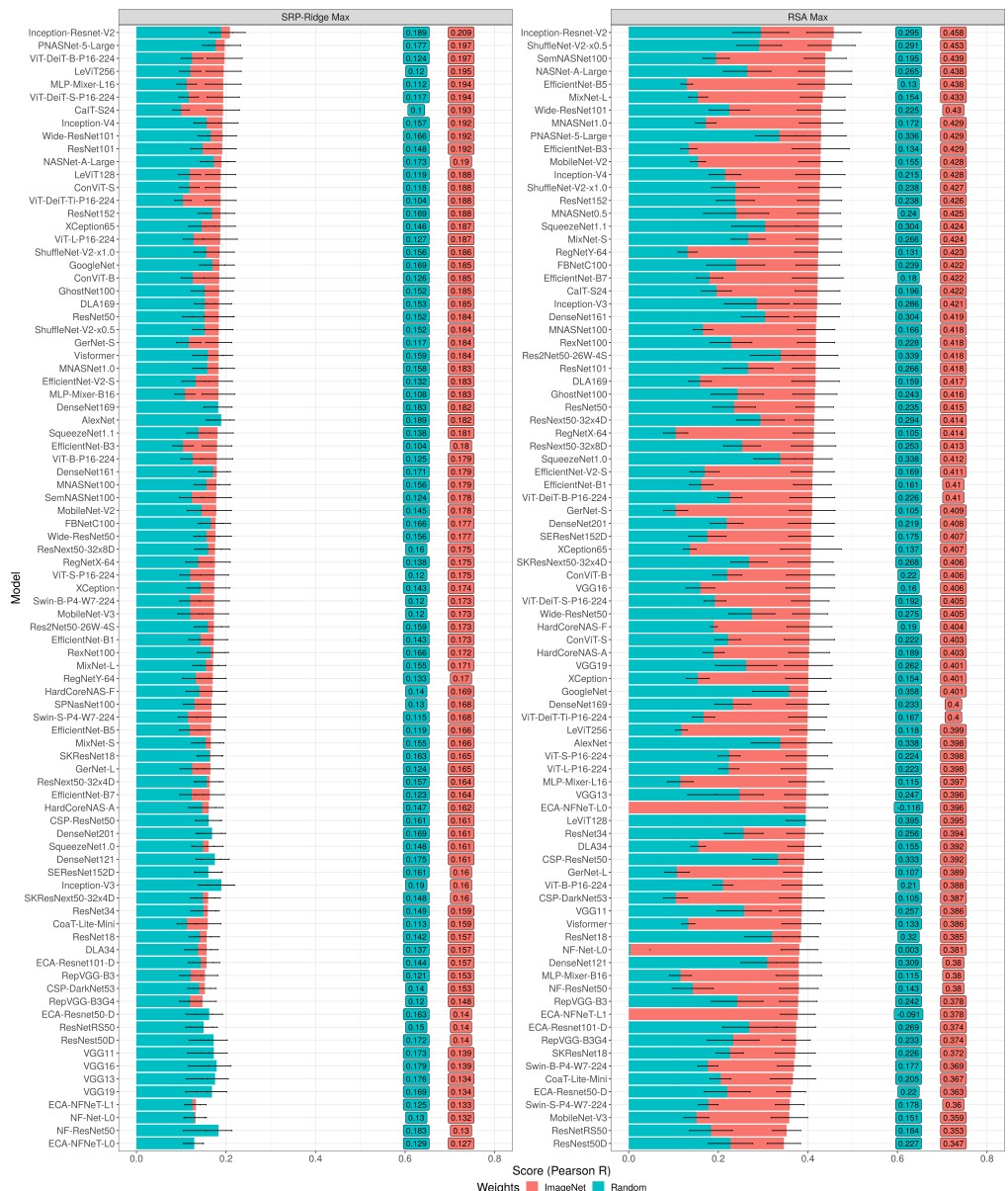

Figure 8: Rankings across model architectures, sorted by the scores of the ImageNet-trained variants (red) of each. Instances in which the randomly initialized variants (blue) outperform their ImageNet-trained counterparts are visible in those rows where the blue entirely overlaps the red. Error bars are 95% bootstrapped confidence intervals across the 6 cortical areas.

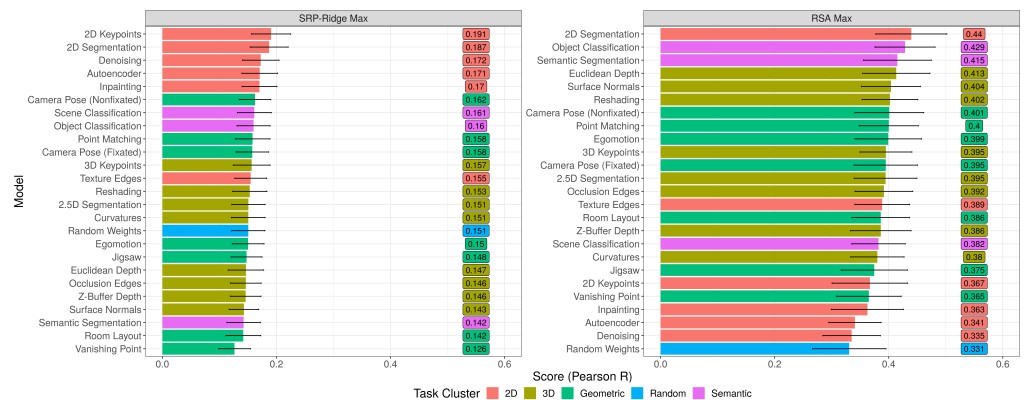

Figure 9: Rankings across Taskonomy encoders, color coded by the Taskonomic cluster to which each belongs. Notice the contrast between the SRP-Ridge Max (which favors 2D tasks) and RSA Max metric (which favors Semantic tasks), but also the relative rank of 2D Segmentation in both.

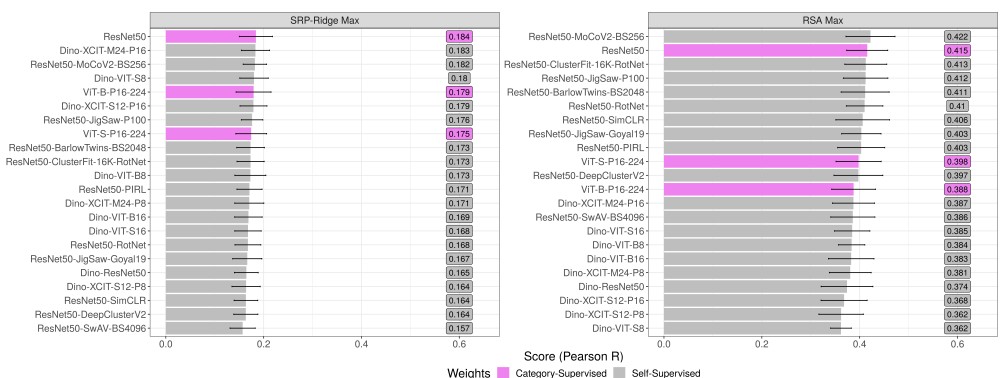

Figure 10: Rankings across self-supervised models. Where available, the category-supervised version of a given model architecture is shown in violet.

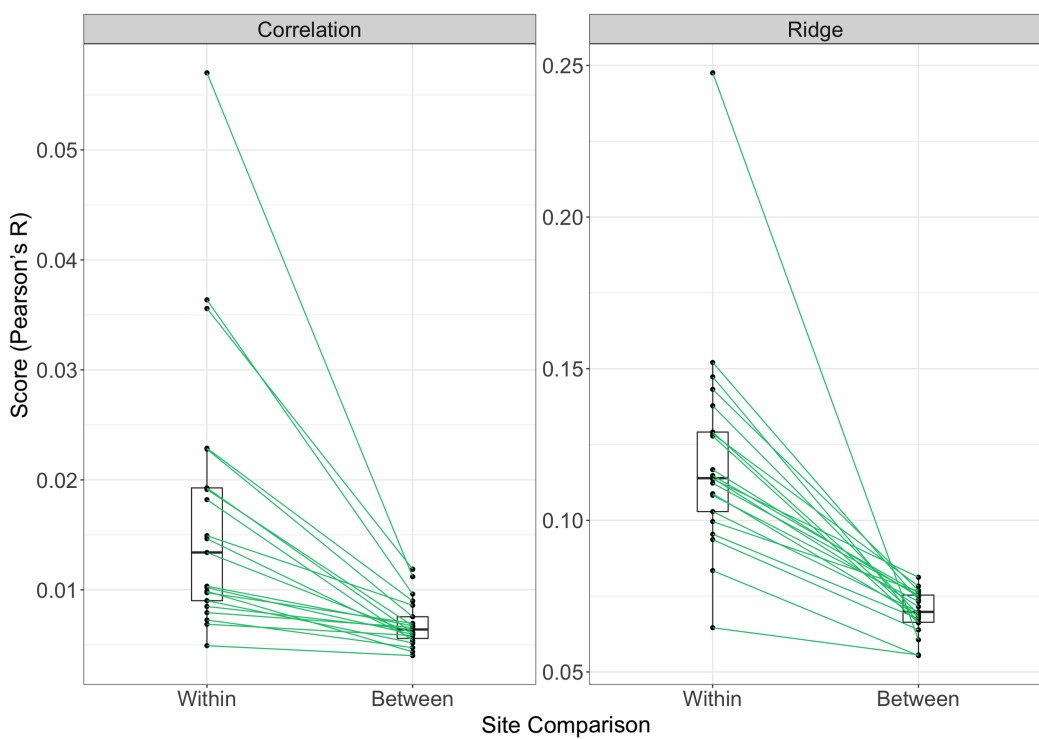

Figure 11: Differences in the predictive power of neurons predicting other neurons *within* site versus neurons predicting other neurons between site. Each paired set of points in this plot is a distinct cortical site (cortical area + cortical layer – 21 in total). The Correlation metric is the average of up to 1000 pairwise comparisons between cells from within the same cite and cells from other sites. The Ridge metric is the iterative prediction of individual neurons using up to 1000 neurons from within the site versus 1000 neurons from other sites. (The number of predictors was subsampled to ensure an equal number of neurons contributed to the within versus between samples). What this plot demonstrates is that neuroanatomical regions do in this case meaningfully correspond to regions with distinct representational profiles, a fundamental component of functional specialization.)

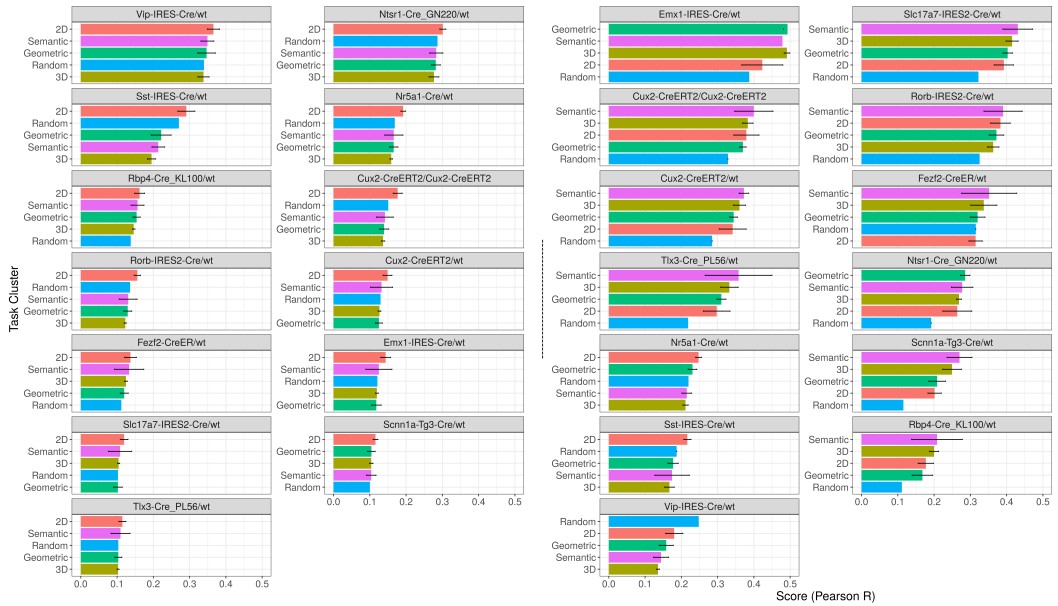

Figure 12: Taskonomy scores across genetic cre line. On the left is SRP-Ridge Max metric; on the right is the RSA Max metric. The facets are shown in descending order by the overall magnitude of their mean scores. Of note: Large-scale motifs present when aggregating across cortical area (such as the dominance of 2D models in SRP-Ridge; the dominance of semantic models in the RSA Max; and the lack of clear Taskonomic dissociations) are recapitulated across cre line. Nevertheless, some differences are salient. For example, the aggregate scores for certain cre lines using the SRP-Ridge Max metric are much higher on average than the scores obtained in any cortical area.

| Area | Structure | Abbreivation | Neuron Count | Proportion |
|------|-----------|--------------|--------------|------------|
| V1 | Primary Visual Area | VISp | 3150 | 0.473 |
| LM | Lateral Visual Area | VISl | 2051 | 0.304 |
| RL | Rostrolateral Visual Area | VISrl | 64 | 0.01 |
| AL | Anterolateral Visual Area | VISal | 780 | 0.115 |
| AM | Anteromedial Visual Area | VISam | 189 | 0.026 |
| PM | Posteromedial Visual Area | VISpm | 493 | 0.073 |

Figure 13: A glossary of areas in the mouse visual cortex.

| Task | Cluster | Definition |
|---|---|---|
| Autoencoder | 2D | Image compression and decompression |
| Object Classification | Semantic | 1000-way object classification (knowledge distillation from ImageNet). |
| Scene Classification | Semantic | Scene Classification (knowledge distillation from MIT Places). |
| Curvatures | 3D | Magnitude of 3D principal curvatures |
| Denoising | Other | Uncorrupted version of corrupted image. |
| Euclidean Depth | 3D | Depth estimation |
| Z-Buffer Depth | 3D | Depth estimation. |
| Occlusion Edges | 3D | Edges which include parts of the scene. |
| Texture Edges | 2D | Edges computed from RGB only (texture edges). |
| Egomotion | Geometric | Odometry (camera poses) given three input images. |
| Camera Pose (Fixated) | Geometric | Relative camera pose with matching optical centers. |
| Inpainting | 2D | Filling in masked center of image. |
| Jigsaw | Geometric | Putting scrambled image pieces back together. |
| 2D Keypoints | 2D | Keypoint estimation from RGB-only (texture features). |
| 3D Keypoints | 3D | 3D Keypoint estimation from underlying scene 3D. |
| Camera Pose (Nonfixated) | Geometric | Relative camera pose with distinct optical centers. |
| Surface Normals | Other | Pixel-wise surface normals. |
| Point Matching | Geometric | Classifying if centers of two images match or not. |
| Reshading | 3D | Reshading with new lighting placed at camera location. |
| Room Layout | Geometric | Orientation and aspect ratio of cubic room layout. |
| Semantic Segmentation | Semantic | Pixel-wise semantic labeling (via knowledge distillation from MS COCO). |
| Unsupervised 2.5D Segmentation | 3D | Segmentation (graph cut) on RGB-D-Normals-Curvature image. |
| Unsupervised 2D Segmentation | 2D | Segmentation (graph cut) on RGB. |
| Vanishing Point | Geometric | Three Manhattan-world vanishing points. |

Figure 14: Task definitions and affinity clusters provided by Taskonomy [53].