# OpenReview forum: "Neural Regression, Representational Similarity, Model Zoology & Neural Taskonomy at Scale in Rodent Visual Cortex"
_NeurIPS.cc/2021/Conference — NeurIPS 2021 Poster_

### Official Review · Reviewer_HWeU · 2021-06-29

**Rating:** 8
**Confidence:** 4

**Summary:**

The authors present a large-scale empirical study comparing artificial neural networks against mouse brain recordings using correlation-based metrics. They utilize a variety of publicly available neural networks as well as a large brain data corpus provided by the Allen Brain Institute. Their metrics are sampled from the field and rely on linear correlation based comparisons. They do also provide a variant metric that is more performant than the recently proposed BrainScore. They demonstrate that an artificial network’s training task and architecture individually influence metric performance. They also replicate some analyses previously done with primate data to emphasize similarities and differences.

**Limitations And Societal Impact:**

Yes, although see my comment #3 above.

**Main Review:**

*Strengths:*
The study is timely and comprehensive. It provides a sufficient level of rigor to establish itself as a viable benchmark for relating mouse visual neuron recordings to artificial network activations. Overall it is well written with easy-to-parse plots. The appendix also provides a concise and convincing defense for the methods chosen in the main manuscript.

*Weaknesses:*
The claimed contribution of the work is to better understand what does/doesn't matter when comparing deep neural network activations against mouse brain activations. The abstract offers some argument for why we would want to do such a comparison in the first place, but in general the paper is pretty well fixated on the opening question of how well the networks fare as models of mouse cortex instead of what does comparing network responses to mouse cortex tell us. I realize that this work is building on an ongoing research thread from a couple of labs, who all provide their own motivations. Nonetheless, I still find myself wondering what new information the results gives us regarding biological & artificial networks. Maybe their emphasis on equivalences between primate and mouse is key? My recommendation is for them to try to make this point of *why the comparison matters in the first place* more salient in the intro or discussion.

The references need to be reviewed carefully. There are many [10, 26-28, 30-34, 36-42, 44-48] that do not include a publication venue. Others [24, 51, 74] are listed as arxiv preprints, but have since been published at peer-reviewed venues. Finally, [53] and [56] appear to be the same citation.

*Additional minor comments:*

1) Assuming you have this sort of resolution in your dataset, how well do the non-neural network baselines & neural networks perform on layer 4 primary cortex data? It has been suggested that this region has a dearth of “complex cells” as compared to primate data {1}. So I would imagine that the Gabor bank in particular will perform comparatively well in this area. I am asking because the Gabor filter bank model is still widely accepted as the standard among “interpretable” methods {2}, and given your discussion at the end about the meager overall performance it would be interesting to see if this focused area has a higher ceiling. Maybe this question is answered with the cre line split, but I’m not familiar with the different lines.

2) While I realize this is far from the authors’ problem to solve, I take issue with the phrasing of correlational scores as a “geometric” analysis. For example, one of a few quotes from the paper: “The resultant coefficient constitutes the score for how well a given model layer predicts the representational geometry of a given cortical area.” I found no mention of geometry in [56]. A quick internet search got me to {3}, which says “The dissimilarities can be interpreted as distances in the multivariate response space. The RDM thus describes the geometry of the arrangement of patterns in this space.” Maybe that’s where they got the term from? I find that it is a stretch to relate correlation values to distances and then further to a geometric description. My recommendation to the authors: use the term if you want, but I would suggest that you at least specify exactly what you mean by it. You should probably also differentiate your definition from others, e.g. that used by {4}, {5}, and {6}.

3) It appears to me that the overall differentiation among the models is quite small. This could lead to a possible alternative conclusion that none of these differentiations (task, architecture, depth, etc) really matter. Or, alternatively, that the metrics themselves are not useful for this type of differentiation. The noted winners/losers among architectures & tasks are certainly interesting observations, but I think it would help balance the paper if the authors commented on the general lack of variation in scores reported from figures 1 and 2. This would go well next to their already helpful discussion on the lack of total performance at the end of the paper.


{1} https://www.jneurosci.org/content/jneuro/28/30/7520.full.pdf

{2} https://www.annualreviews.org/doi/abs/10.1146/annurev-vision-091718-014731

{3} https://journals.plos.org/ploscompbiol/article?id=10.1371/journal.pcbi.1003553

{4} https://www.nature.com/articles/s41586-019-1346-5

{5} https://elifesciences.org/articles/44526

{6} https://www.sciencedirect.com/science/article/pii/S0042698915003600

**Time Spent Reviewing:**

6

---

> ### Author Response · Authors · 2021-08-10
> **Response to Reviewer HWeU**
>
> We would like to thank the reviewer for their detailed feedback, encouragement and estimation of our work. We hope the responses below will help ease any lingering uncertainties! We've limited our comments to weaknesses, and have attempted to respond point by point.
>
> The reviewer's main comment regarding motivation is one we take to heart. For the sake of discussion, and perhaps for distilling a more titrated version of this in the final paper, we offer three points of justification below:
>
> 1. Machine vision models appear to map quite well to recordings from the primate visual cortex. Yet they don’t seem to explain mouse visual cortex to anywhere near the same degree. Rodents and primates diverged around 80-100 million years ago, yet both rodents [1, 2] and primates recognize objects (the common ancestor of the two likely did the same), and so must have solutions that approximate something like the object recognition we attempt in computer vision. If our models match primate visual areas and not mouse visual areas, are we overfitting our architectures to an artifact? Perhaps object recognition is what the two species have in common and we are missing something fundamental about object recognition architectures.
>
> 2. Related to the above, perhaps mice simply have a different solution to common visual tasks. Maybe one that requires far fewer resources and far less depth, and one which does not appear to map neatly onto current networks. There is no reason to believe that there is a single solution to how animals or machines should see. Current research in computer vision is pushing the boundaries of what is possible from our understanding of the primate cortex. Other solutions may uncover new computational principles or efficient architectures (which with the extreme computational demands of modern computer vision we are in dire need of).
>
> 3. A recurring theme in both AI and psychology is that humans are not aware of the wide range of complex tasks we solve, never mind how we solve those tasks. The computer vision community is very focused on tasks which seem intuitive to us: object recognition, segmentation, et cetera -- roughly the tasks represented in Taskonomy. We have no reason to believe this is an exhaustive list of tasks that humans perform, and far less reason to believe that these are the relevant tasks to other animals like mice. These sorts of comparisons point out that either our architectures are fundamentally different from the mouse visual cortex (the case described above, where we may discover important new principles) or our tasks are lacking. All of our investigations of vision (machine or animal) are limited by the fact that we don’t yet understand exactly what vision is: the dark matter of vision are the missing visual tasks that we must be performing in order to create the rich visual experience that we inhabit as humans. Other animals, particularly those for whom object recognition is not so overwhelmingly central, may help uncover these hidden tasks, that may in turn shed light on primate vision and on the core underlying algorithms of vision itself.
>
> Additional Minor Comments:
>
> 1. [Gabors in Primary Visual Cortex, Layer 4] Per the reviewer’s suggestion, we further subdivided the results of the neural regression with Gabor filters into cortical area + layer. While Gabor filter scores in VISp, layer 4 were overall pretty low (at 0.057), scores in VISp, layer 6 were the highest of any cortical area + layer (at 0.167) -- competitive even with scores from some of the Taskonomy encoders and 26% higher than the next best cortical area + layer (VISam, layer23 at 0.130).
>
> 2. [Representational Geometry] We thank the reviewer for pointing out the potential misunderstanding with the use of the term representational geometry -- which we now realize may be a term of art in the human neuroimaging literature that does not translate easily elsewhere. To avoid misunderstanding, we can replace this term simply with representational similarity, thereby circumventing the other connotations of geometry that do not transfer across fields.
>
> 3. [Non-Differentiation Among Models] This is an important point, and a good suggestion. The non-differentiation among models, at least as we’ve interpreted it, underscores again that the typical design variation across deep net models (especially in terms of architecture, but also in terms of datasets collected by humans and overly saturated with anthropocentric labels) does not translate to meaningful variation in the prediction of neural activity in mouse visual cortex. This could allow us to reiterate a larger thematic point about the necessity of ethologically relevant objective functions and environmentally plausible datasets / augmentations (based on how the mouse actually behaves in its environment, and how those environments impinge on the sensory interface). We will try (space permitting) to update the conversation on variance explained with this additional point.
>
> [Citation Issues] We thank the reviewer for their assistance in updating the citations. We’ve now tracked down those rxiv papers since published in journals / conferences, added publication venues and attended to the redundancies where they exist.
>
> [Citations: Updated Venues]: ([10] NeurIPS 2019 Workshop Neuro AI, [26] https://arxiv.org/abs/2004.10934, [27] AAAI'17, [28] Proceedings of ICML, 2019, [30] ICML 2019, [31] CVPR 2019, [32] BMVC 2019, [33] ICLR 2021, [34] CVPR 2019, [36] CVPR 2018, [37] https://arxiv.org/abs/2103.14030, [38] CVPR 2017, [39] ICML 2021, [40] ICML 2021, [41] https://arxiv.org/abs/2104.06399, [42] CVPR 2020, [44] https://arxiv.org/abs/2105.01601, [45] ICCV 2019, [46] ICML 2021, [47] ICLR 2021, [48] CVPR 2021)
>
> [Citations: Arxiv, Now Published] ([24] BMVC 2016, [51] CORL 2019, [74] ICLR 2020)
>
> [1] Zoccolan D, Oertelt N, DiCarlo JJ, Cox DD. A rodent model for the study of invariant visual object recognition. Proceedings of the National Academy of Sciences. 2009 May 26;106(21):8748-53. https://www.pnas.org/content/106/21/8748.full
> [2] Broschard MB, Kim J, Love BC, Freeman JH. Category learning in rodents using touchscreen‐based tasks. Genes, Brain and Behavior. 2021 Jan;20(1):e12665.
> https://onlinelibrary.wiley.com/doi/full/10.1111/gbb.12665

---

> > ### Comment · Reviewer_HWeU · 2021-08-30
> > **Increase score**
> >
> > Apologies for the delay in this reply -- I thought I had sent it some time ago.
> >
> > *Summary:*
> > Given the author's responses to my own as well as the other reviews, I am further convinced that this paper is of high enough quality to be published at NeurIPS. I have adjusted my score accordingly. My primary recommendation for their camera-ready revision is to focus on messaging regarding what their contributions are -- this is a reoccurring issue with the reviewers.
> >
> > *Detail:*
> > I find the motivations provided to be relevant and significant. The line of discussion in points 1 & 2, regarding the divergence (or convergence) of solutions between mice and monkeys, seems particularly salient. In an attempt to relate it back to your work, you say in your abstract that previous work has suggested that DNN responses do not correlate well to mouse cortex. My interpretation is that previous work suggested that this would imply divergent solutions, where the monkey has evolved a solution that is more similar to DNNs. I believe you are suggesting in your work that the DNNs can be used to explain activity in the mouse cortex, at least to some degree. You certainly don’t answer points 1 or 2, but it seems to me like you provide evidence supporting the hypothesis of a shared evolved solution between these animal models, which has enough overlap with DNNs to appear in correlational measures.
> >
> > It is my opinion that adjusting your introduction to include a broader perspective along the lines you suggested, as well as the response about a guiding hypothesis given to Reviewer ezgN, would make the paper more intriguing for an uninitiated reader. Perhaps more importantly, being explicit and concise about your contributions will make the paper easier to contextualize for the veteran reader. You list a number of contributions in the abstract and introduction, including a novel regression model that you have since agreed to revise in response to reviewer 5VYT. Tweaking existing ML solutions to achieve higher scores is common as a primary contribution in ML papers, but I do not think this paper needs to rest on that as a significant contribution. In my opinion the improved regression scores underly a bigger point, which is that you provide evidence that 1) contradicts the current zeitgeist of the field and 2) clarifies finer details about task specificity and architecture goodness-of-fit. I trust that you will take care to balance the trade-off between making overly grandiose claims about scientific significance and making the larger motivation and relevance to (biological & artificial) intelligence clear.
> >
> > The result with the Gabor filters is quite interesting to me, especially considering the strong direct thalamocortical connectivity (both inputs and outputs) for layer 6. An increased correlation with Gabor filter responses would imply that the neurons are more linear, I guess. For me at least, your additional study raises more questions than it answers. However, I realize that this is not the focus of your paper. I think adding this context will help the more classically trained reader relate DNN results back to traditional work with Gabor filters. If you can’t fit it, then I would suggest adding it to the appendix with a brief reference in the main text.
> >
> > I appreciate your responses to my other concerns, and trust that you will successfully incorporate them into the final submission.

---

### Official Review · Reviewer_5VYT · 2021-07-17

**Rating:** 6
**Confidence:** 4

**Summary:**

Several other reviewers have expressed that they find the current scientific contributions of the paper significant enough, so I will increase my score to a 6, even though the authors did not quite address my major concern about the novelty and utility of the proposed neural regression (i.e. what is the empirical benefit of neural regression over just ridge regression? if that is not understood, the authors should disclose that and not put so much emphasis on the importance of the regression method (in the title, in the abstract, and in the main text)).

I believe that the paper can benefit from reducing the amount of time spent discussing the neural regression method and the RSA vs encoding model comparisons, and more time focusing on the actually novel and significant part which is the comparison of the findings in mice to the previously established findings in monkeys and humans. And just to clarify, I think that only discussing what the findings are in mice is not enough on its own, because all the analyses that the authors carry out have been previously done in other organisms. So to actually have a significant contribution, I think it's important to emphasize the comparisons of the results in mice to the previously established results in other organisms.

=============== post-rebuttal =================

The manuscript presents an exploratory empirical study of the relationships between the mouse visual system and representations extracted from computer vision neural network models. The authors investigate representations from a large number of computer vision models, as well as representations from the same models but when the models are initialized randomly. They also explore two ways of relating the mouse recordings to the representations from the computer vision models: encoding models and representational similarity analysis. The authors additionally propose what they claim is a new encoding model methodology, though I remain unconvinced of its utility.

**Limitations And Societal Impact:**

Yes

**Main Review:**

The manuscript was easy to read and provides a broad exploration of an interesting question. I do however have several concerns about this work. I've summarized these below.

1. Methodological concerns. The authors claim to propose a "novel, highly optimized neural regression method" for encoding models. However I'm unclear about what parts of the proposed method (sparse random projection + ridge regression via generalized cross validation) the authors are claiming to be novel, and I'm also unclear about which of those parts are actually leading to a boost in prediction performance. The authors claim that ridge regression is a less popular encoding model method, though it is in fact the default method in encoding models in the language literature (Wehbe et al 2014, Huth et al. 2016, Jain and Huth 2018, Toneva and Wehbe 2019,...). The authors use leave one out cross validation which is indeed an uncommon choice, but it's also the default setting on the RidgeCV method in sklearn, so it's unclear what is "highly optimized" about this method. Also, it would be helpful if the authors could provide some clarification (both in the rebuttal and in the paper) about exactly what samples are used to train the encoding models (both in the mouse and the monkey experiments). Were separate encoding models built for different individuals? If separate encoding models were built, it's not clear how the predictions or predictions performance is aggregated across individuals. How were the repetitions of different image stimuli handled in the leave one out CV setting? I imagine that leaving out the data corresponding to one image while keeping data that corresponds to other repetitions in the training set can lead to some unfair underestimate of the generalization error. It would be interesting to compare the zero-shot performance of the leave-one-out proposed method to the zero-shot performance of the default Brain Score encoding model (i.e. zero-shot meaning that all repetitions of a test image are removed from the training set).

2. Questions about some of the interpretations.
- L214-220: The authors interpret the result that transformer-based architectures show the largest differences in prediction performance between the trained and randomly initialized computer vision models as evidence that some of the better performing randomly initialized models may be due to an underlying convolutional architecture. Another possibility is that the randomly initialized models predict the neural recordings well across architectures, but the trained transformer-based architectures are much better at predicting the neural recordings. This possibility is consistent with a large difference in prediction performance between the trained and randomly initialized transformer-based models. To disentangle these, it would be helpful if the authors compared the prediction performances across different randomly initialized architectures with a statistical test.
- The authors compare the results from RSA with those of encoding models and state that "with more probe images, we expect that the parametric linear mapping would be more competitive with the nonparameteric distances of representational similarity". It's not clear to me that the correlations from encoding models being lower than those from RSA means that the encoding model results are less "competitive". The results from an encoding model are from generalizing to a previously unseen datapoint, and the RSA results are not. It would be helpful if the authors provide an estimate of the chance performance in both the encoding model and the RSA setting to show that one is not inherently easier than the other.

3. Some lack of clarity in the methodology and experimental settings. The manuscript has a clear exposition for the most part but there are some methodologically details that need to be better explained. Some examples:
- The questions that I brought up in the previous point
- L92-98: what is being clustered and how? Are these clusters from the original Taskonomy paper or did the authors actually obtain these "purely data-driven clusters".
- L111-120: for a reader who is unfamiliar with RSA, the description in these lines may not be easy to parse. For example "We compute RDMs by correlating the activations of all neurons in a given neural site to the 118 images in the stimulus set" sounds like the authors are correlating neural activity to images, instead of computing pairwise correlations across the neural activations. Here it is again not clear how separate subjects are handled.
- L185-196: are these results using the mouse or the monkey data?

**Time Spent Reviewing:**

4

---

> ### Author Response · Authors · 2021-08-10
> **Response for Reviewer 5VYT**
>
> We thank the reviewer for their time and pointed feedback. To the best of our ability, we’ve attempted to respond to the reviewer’s concerns, point by point, below.
>
> 1. Methodological Concerns
>
> [Neural Regression Method] The reviewer is correct in pointing out that none of the subcomponents of the neural regression method are new, and also that many involve simple defaults in standard python libraries (including the RidgeCV function). What we were referring to as “novel” was simply the combination of these parts into a single pipeline; what we meant by “highly optimized” were the gains on performance and speed instantiated by their combination. In retrospect, we see both of these terms were misleading, if not incorrect, and will rephrase the statement in the abstract (and subsequent similar phrasings) as: “We introduce a more efficient, more accurate neural regression pipeline that achieves substantially higher scores than previous methods on the publicly available benchmarks of primate BrainScore.”
>
> [Clarification on Encoding Models] The reviewer draws due attention to a few areas of underspecification in the neural regression procedure. The first item we should clarify is the preprocessing of the neural data. Prior to processing, we have for each neuron a response vector of size [119 x 50], corresponding to the 119 images and the 50 presentations of each image. We then average across presentations to give us a response vector of size [119 x 1] for each neuron. We then fit encoding models individually to each neuron in the dataset, calculating the predicted mean activity per image with the GCV (leave-one-out) prediction [119 predictions in total]. After calculating regressions per neuron, we had various options as to how we might aggregate. After running various analyses comparing the reliability of neural response in various cortical areas across mice and finding little discernible difference with respect to individual donors, we decided to aggregate by cortical area only, ignoring the provenance of a given cell (i.e. its donor). To summarize, we enumerate our answers the reviewer’s questions as follows:
> - Models are fit per neuron and the performance is then aggregated across cortical area, irrespective of their donor mouse.
> - We never use the same image (regardless of repetition) in both the training and test set. Each image is held out during the cross-validation procedure in such a way that no information about that image remains in the training set.
>  - If we’ve understood the reviewer correctly, the zero-shot evaluation (the only generalizable evaluation) is precisely how well we evaluate the performance of our encoding models.
>
> 2. Interpretations
>
> - [Interpretation of Transformer / Convolutional] If we’ve understood the reviewer’s alternative explanation of the results indicating an architectural inductive bias for convolutions correctly (that random initialization may perform equally well for transformers and convolutional models), and his suggestion for a statistical test, then the test that best arbitrates on this question is a t-test between the randomly initialized convolutional and randomly initialized transformer architectures. This test shows that randomly initialized convolutional architectures dramatically outperform randomly initialized transformer architectures (p = 1.63e-14, Hedge’s g = 2.29). This suggests the inductive biases inherent to convolutional architectures may provide the basis for their better predictive power.
>
> - [Chance Performance / Comparing RSA - SRP-Ridge] We agree on this front. Other reviewers have mentioned the disjoint when comparing the scores from the RSA to the SRP-Ridge, and we have decided to remove the statement accordingly. In terms of chance performance, though, we were inspired to take an AlexNet architecture and calculate permutation tests for our RSA Max and SRP-Ridge Max metrics, permuting across the image dimension in each feature map, before performing the rest of the analysis (and repeating this 1000 times). Strikingly, the mean scores from both sets of permutation tests suggested chance was effectively zero in both cases: with mean scores [+/- confidence intervals] of 0.0001 [-0.0175, 0.013] for RSA Max and
>
> - [More Methodological Details]
>      - What is being clustered are 24 different taskonomy models according to how well one model transfer learns to another. The clusters we use are taken directly from the original Taskonomy paper (Figure 13): http://taskonomy.stanford.edu/taskonomy_CVPR2018.pdf
>      - Lines 111-120: Duly noted. We will edit this to read: “RDMs were calculated by computing the pairwise correlation coefficients between the neural response vectors for each image...” As with the neural regression, RDMs were aggregated only by cortical area (with pairwise comparisons across all the cells in that area regardless of donor), totaling 6 RDMs.
>      - Lines 185-189: All results beginning with Sec 3.2 (including these) pertain only to the mice.
>
> [1] Golub GH, Heath M, Wahba G. Generalized cross-validation as a method for choosing a good ridge parameter. Technometrics. 1979 May 1;21(2):215-23. https://www.jstor.org/stable/1268518

---

### Official Review · Reviewer_omuc · 2021-07-23

**Rating:** 6
**Confidence:** 4

**Summary:**

Extracts intermediate features (across all layers) for models pretrained on ImageNet classification (and other computer vision tasks) and tests its modeling performance of mice two-photon neural responses (from 6 cortical areas) using regression and RSA. Features at intermediate depths are better, there is a slight hierarchy between neural network depth and cortical area, and there seems to be no relation between the pretraining task and the cortical area the trained network models best.

**Limitations And Societal Impact:**

Biological datasets are always limited, though not directly pointed out, it is understood. Discussion about societal impact of a better understanding of the visual system is not directly provided, though I would think it may be unnecessary.

**Main Review:**

Originality
Though techniques used are standard, the extent of the experiments and the animal model (mice) is novel and significant.

Quality
Experiments are sound and enough information is provided to allow replicability.

The main drawback of the paper in its current form is the figures, though they contain information to draw the conclusions drawn in the paper, it is not easy to see. Fig 1, for instance shows performance metrics of pretrained vs randomly initialized models but the models have been sampled (from the total of 110), the red bar goes below the blue bar if the value is lower and the error bars overlap so it is hard to see where they start or end. Perhaps a two-sided violin plot per model (instead of the bars) could show the distribution for blue and red better. Or maybe a scatter plot (x axis trained, y axis random) where each point is a model could show all models and more easily show which version is better (in comparison to the diagonal), error bars could also be included per point or perhaps omitted for clarity, and important points (i.e., models) can be labelled (with colors or marker styles or text above the point). All results from Sec 3.3 (other than the best network being Inception-ResnetV2) are close to impossible to see in Fig. 1. In Fig. 2, perhaps adding a table with the averages per task group will help show the trends and differences between the two metrics (as done for figure 3). Fig. 3 could have also been a table (or perhaps two), it will be easier to read the numbers and gauge the difference that way (it is also currently unclear what is your assumed cortical hierarchy and task hierarchy). Given the amount of work put into obtaining these good results, spending time finding a better way to convey them might be worthy and will surely make the paper more attractive/citable.

I do not think the method comparison with BrainScore regression (Sec 3.1) adds to this paper (and it may take away from it). I understand this was asked of you before but it does not actually show that results from this paper (in mice) will hold if using the BrainScore regression method, it only shows that both methods give similar regression results on the same data (which is expected given that the changes are minor). That the method transfers to primate data (with the very different visual system and recording methods) might be interesting but I presume so will many other methods (including BrainScore’s) and regression performance or transferability across animal models is not the focus of this paper so it feels disconnected. It may be worth considering bumping it to the end of the result section (as an extra transfer result) or even to the appendix.

Clarity
Well written, though sometimes wordy; for instance, the main results of the paper at the end of the introduction could be a lot bolder/more direct even if obviating some details (for instance, 1 → “Training matters: features from trained models predict/model neural responses better than features from randomly initialized models.” or phrases like “features learned in service of some task” → “learned features” or “there may be a slight upwards gradient in complexity” → “complexity increases”).  Of course, the work and results still hold but I think ML readers will appreciate a less hedged writing style.

For reproducibility, can you add a sentence about how images were modified to work as ImageNet images, how do you deal with the aspect ratio difference (if any), sizing and color.

Perhaps it is worth providing the proportion of cells (out of the 6.6K) from each cortical area (and the name of the areas). A single sentence should do. “Dataset areas are divided as V1 (50%), AM (10%),  PM(5%), ….” (could be in supplementary).

For the Taskonomy models, you say “we choose to extract representations across all layers”(l.90), to confirm, this means through all encoder+decoder layers (i.e., different tasks may have different depths)?

I am confused about your ridge regression, do you cross-validate or not? because it says you use “a lambda penalty of 1.0” (l.132) but then in the same line you say you “monopolize generalized cross-validation” (?), I presume you do use LOOCV (if so, I would drop the sentence about using a lambda of 1 and perhaps provide the lambda grid you searched over).

Also, to confirm, once you cross-validate the penalty, you use all the data to fit the final linear regressor (with the chosen lambda), right? if so, the regression correlation may be optimistic (i.e., it reports cross-validation results). This is ok as long as what’s reported is clear.

The interpretation in Sec 3.4 seems premature given that it is only truly one task (Semantic segmentation) that moves up from SRP to RSA (and at the same time, scene classification moves down from SRP to RSA). It is “only” a hypothesis so it is up to the authors to decide whether they are ok with it.

Changes in performance in Sec. 3.6 are so small I do not think one can even say they point in any direction (l.251).

Results from Sec 3.8 are neat, I thought the lack of functional specialization across different cortical areas in mice was well known; maybe you can find some references for it and frame your results as further confirmation. I would also reference some of the discussions about cortical area hierarchies.

In l. 296, pointing regression scores are lower than RSA scores seems unimportant. They are both correlation scores but in very different spaces.

Great discussion.

Other minor remarks

- the word “later” in l.33 is odd.
- you end up using 6619 cells (not 59K as claimed in the abstract)
- in l. 132, it will be better to name scikit-learn [52] rather than just the citation or “implemented with a popular machine learning toolkit [57]” if you don’t want to name it.
- in l. 152, drop “In this work,”
- i would reference the figures before the written results (i.e., closer to the top of each 3.x subsection rather than at the very end), for instance, the line in 195-196 could go up to l.184
- l.219 ‘writ large”?
- Reference to fig 4 (l.257) appears before reference to fig 3 (l.281), fig 3 should probably be fig 4 and viceversa.
- For 3.x sections, perhaps consider putting result directly in the titles, for instance, section 3.5: "how well do non-neural network baselines predict rodent visual cortex" -> "non-neural network baselines fail to predict rodent visual cortex”
- Showing the neural RDMs for different areas and perhaps some for a selected layer in a trained model could be interesting (perhaps in appendix)

Significance
It fills a research gap in visual neuroscience and will be a good reference for future studies.

Overall
This is a good paper that is on the verge of acceptance. With some improvements I would have no problem accepting it.


Update:
I have bumped my score from 5 to 6 after the authors addressed some of my concerns and those of other reviewers.

**Time Spent Reviewing:**

7

---

> ### Author Response · Authors · 2021-08-10
> **Response to Reviewer omuc**
>
> Reviewer omuc
>
> We thank the reviewer for their encouraging, extensive and highly detailed feedback. We hope the improvements offered are sufficient, and are happy to adjust further over the course of discussion! We’ve responded point by point below.
>
> [Figures] A point very well taken. We did admittedly struggle to convey all the information we hoped to convey in the space allotted. That being said, we’ve now added to our anonymous Github repository a number of the figures suggested by the reviewer, a few of which we consider significant improvements or candidates for inclusion in the main paper:
> - scatterplot of ImageNet versus Random scores, with points of interest labeled.
>     - https://github.com/annonymousecode/Model-Zoology-and-Neural-Taskonomy/blob/master/rebuttal_figures/imagenet_random_scatterplot.png
> - a paired spaghetti plot showing the distributions of Random versus ImageNet scores, with statistics and points of interest labeled.
>     - https://github.com/annonymousecode/Model-Zoology-and-Neural-Taskonomy/blob/master/rebuttal_figures/imagenet_random_pairplots.png
> - a figure that more clearly shows the hierarchy from VISp onwards, with nonparametric omnibus and pairwise statistics:
>     - https://github.com/annonymousecode/Model-Zoology-and-Neural-Taskonomy/blob/master/rebuttal_figures/heirarchy_plots.png
> - a figure that demonstrates how the rankings of task clusters (e.g. 2D, Semantic) shift between the RSA and SRP-Ridge metrics. (This could be stitched with Figure 2, or added to the appendix).
>     - https://github.com/annonymousecode/Model-Zoology-and-Neural-Taskonomy/blob/master/rebuttal_figures/task_cluster_plots.png
>
> - a table that recapitulates the data of figure 3.
>      - https://github.com/annonymousecode/Model-Zoology-and-Neural-Taskonomy/blob/master/rebuttal_figures/task_cluster_by_area_table.png
>
> [Assumed Cortical Hierarchy] An excellent question. A slight oversight on our part, we omitted reference to the hierarchy we were assuming in describing fig. 3. That hierarchy is based on works [1,2,3], especially Figure 2C-D in [2] and Figure 3C-E-F in [3]. The proposed hierarchy consists of a ventral / dorsal split after primary visual cortex (VISp), with VISp - VISl in the ventral stream and VISp - VISal - VISrl - VISpm - VISam in the dorsal stream. One figure (rebuttal_figures/hiearchy_plots.png) we’ve added more clearly shows this hierarchy (split into ventral and dorsal streams) in the style of Figure 3C in [3].
>
> [BrainScore Results Misplaced] Duly noted. As the reviewer says, the centering of this result was a response to previous skepticism of the neural regression method, and was meant only to demonstrate that lower scores in mouse brain (when produced) were not a fault of the mapping. If this no longer seems an issue, we could very easily move this result to the Appendix.
>
> [Simplifying Language] Where possible, we will work to decrease hedging and underscore key results, perhaps with bold?
>
> [Reproducibility: Image Transforms] Images for all models were transformed using the pipeline specified by each model’s authors, and implemented with PyTorch. While some transforms varied across pipeline (for example, resizing with bicubic versus bilinear interpolation), the most common set of transforms for ImageNet models was resizing (without cropping) to (224, 224) and normalization to ImageNet mean and standard deviation. Images for the Taskonomy models (as specified in [5] were resized to (256, 256) and normalized in the range of [-1,1].
>
> [Neural Proportions by Area] Cell counts by area were as follows: VISp (n = 3150, 48%); VISl (n = 2051, 0.31%); VISal (n = 780, 12%); VISpm (n = 493, 7%); VISam (n = 189, 3%); VISrl (n = 64, 1%). We will include this in the appendix.
>
> [Taskonomy Layers] The reviewer makes a good point that this was a bit unclear in the paper. We extract features across all layers of the encoder only (meaning equal depths across tasks). We could perform similar analyses across the features of the decoders, but this would have meant ceding the empirical control of architecture, making subsequent results incomparable. (We'll update accordingly).
>
> [GCV / Lambda Penalty] We use the sklearn default lambda of 1.0 across all regressions, and did not perform any sort of hyperparameter search given limited data. We use the (generalized) cross-validation to derive our predicted neural activity for each image, which we then correlate with actual activity. To summarize, the GCV is used for prediction and not for hyperparameter selection. We will try to make this clearer in the final product. Our choice of GCV was simply as a more efficient replacement for leave-one-out cross-validation (which produces the same predicted values).
>
> [Sec 3.4 Which task best predicts rodent cortex?] We are a bit confused by the reviewer’s comments on this section. In Figure 2, almost all of the 2D tasks (in red) slip from the top and middle rankings under SRP-Ridge to the bottom of the rankings under RSA -- a substantial shift. If the reviewer could clarify, hopefully we can determine whether this is still sufficient evidence for the interpretation we offered in this section. This is perhaps made clearer in the rebuttal_figures/task_cluster_plots.png we've included in our Github repo.
>
> [Sec 3.6 Change in Performance] This is a point brought up by other reviewers and we agree. The conclusion should either be offered with heavy caveat, or any mention of advantage eliminated entirely in favor of stating that ‘both task and architecture matter equally’.
>
> [Sec 3.8 - Hierarchy] Inspired by the reviewer’s suggestions, we revisited the literature on functional organization and hierarchy in mouse visual cortex. The debate, as suggested by recent work like [3,6], has recently shifted strongly in favor of there being both a hierarchy and functional specialization across mouse visual cortical areas. Our results (especially from the RDMs) are strongly concordant with the hierarchy reported in the most recent analyses (see [6], Fig 2a.); a brief summary of this hierarchy would be that inputs from V1 first flow through VISl to the ventral stream, and VISal to the dorsal stream (continuing through VISrl to VISpm then VISam), with a slight but significant increase in receptive field size along the way [3,6]. A figure we’ve added adds nonparametric statistics to this visualization, showing both significance differences across area at the group level, and pairwise differences between VISp and later layers in both RSA & SRP-Ridge. We plan on making this a primary figure in the main paper.
>
> [Sec 3.8 - Functional Specialization] That we don’t see Taskonomic differences across cortical area may suggest a lack of functional specialization, but a more likely explanation is that the visual features used to differentiate the cortical areas in recent work (spatiotemporal frequency, receptive field size, sensitivity to orientation) are too low-level to arbitrate between the features of different Taskonomy networks. For example, optic flow seems to be a key feature for distinguishing VISpm from VISal [7], but is likely absent in the features of the Taskonomy models. All this to say that while our results do suggest a relative homogeneity of function across cortical area, this does not mean functional specialization does not exist -- only that tools used to detect that specialization in primates may not transfer as well to mice.
>
> [Line 296 SRP-Ridge versus RSA] Another reviewer has pointed out that the comparison here is probably irrelevant, and without strong basis for comparison. We will remove it from the manuscript.
>
> [Minor Comments] We agree with many of the reviewer’s edits and suggestions, particularly changing section headings to statements instead of questions.
>
> References
>
> [1] Wang Q, Gao E, Burkhalter A. Gateways of ventral and dorsal streams in mouse visual cortex. Journal of Neuroscience. 2011 Feb 2;31(5):1905-18. https://www.jneurosci.org/content/31/5/1905
> [2] Wang Q, Sporns O, Burkhalter A. Network analysis of corticocortical connections reveals ventral and dorsal processing streams in mouse visual cortex. Journal of Neuroscience. 2012 Mar 28;32(13):4386-99. https://www.jneurosci.org/content/32/13/4386
> [3] D’Souza RD, Wang Q, Ji W, Meier AM, Kennedy H, Knoblauch K, Burkhalter A. Canonical and noncanonical features of the mouse visual cortical hierarchy. bioRxiv. 2020 Jan 1. https://www.biorxiv.org/content/10.1101/2020.03.30.016303v1.full
> [4] https://rwightman.github.io/pytorch-image-models/
> [5] ​​https://github.com/alexsax/midlevel-reps
> [6] Siegle JH, Jia X, Durand S, Gale S, Bennett C, Graddis N, Heller G, Ramirez TK, Choi H, Luviano JA, Groblewski PA. Survey of spiking in the mouse visual system reveals functional hierarchy. Nature. 2021 Apr;592(7852):86-92. https://www.nature.com/articles/s41586-020-03171-x
> [7] Andermann ML, Kerlin AM, Roumis DK, Glickfeld LL, Reid RC. Functional specialization of mouse higher visual cortical areas. Neuron. 2011 Dec 22;72(6):1025-39. https://www.sciencedirect.com/science/article/pii/S0896627311010129?via%3Dihub#bib69

---

> > ### Comment · Reviewer_omuc · 2021-09-14
> > **Just acknowledging the author's response**
> >
> > Thank you for your thorough response (I'm aware it was a lengthy review). I will add some final comments about some points in your response
> >
> > [Figures] For the scatterplots (random vs imagenet), having different ranges for x and y axis could be misleading; at the least, it makes it hard to locate the diagonal of the plot and find the general trends. That said, it seems other plots (like the spaghetti plot) may show trends better. The newer hierarchy figures may not provide as clear of a story as the previous 3a one but I think it provides more information and allows the reader to see the data and make their mind about how much they trust it. Same applies to the summarizing image task_cluster_plots.
> >
> > [Reproducibility: Image Transforms] I would consider adding this info in the appendix. Essentially, we resized to 224,224 from the original ?x? pixels and normalized using ImageNet stats.
> >
> > [Sec 3.8 Functional specialization or not] I would perhaps add a couple of lines either in the section or the discussion making clear you are not claiming a lack of functional specialization across areas.
> >
> > One last minor comment, in caption of Fig. 3, "don't" -> do not (or more boldly "are not")

---

### Official Review · Reviewer_ezgN · 2021-08-01

**Rating:** 7
**Confidence:** 4

**Summary:**

This paper proposes a benchmark for the comparison of neural network models of vision to rodent visual cortex. The benchmark compares models with various architectures (including various CNNs, transformers, and other architectures) and one model trained different tasks to a large database of neural recordings from mouse visual cortex. It relies on newly proposed metrics that are validated on a previous benchmark. These metrics evaluate both the fit to individual neurons and the entire population of neurons. Previous results are replicated and explained in the context of a large scale study. The paper provides analysis of the results as well as predictions on the function and structure of the rodent visual cortex areas.



**Limitations And Societal Impact:**

The authors addressed some of the papers limitations. However there are other issues to address, I have split them into major and minor weaknesses:

Major
- It's important for analysis papers to have a clear objective beyond fitting neural data. The paper lacks hypotheses and predictions on brain function that constrain the space of models and tasks, allow for more in-depth analysis which, in turn, gives solid conclusions.
- In brainscore's benchmark, data was recorded from monkeys tasked with object recognition, so evaluating only models trained on classification makes sense. To my understanding, the mice that are recorded in the dataset were only passively viewing images. Thus, it might be more reasonable to, for example, shift the analysis towards unsupervised/self-supervised objective functions and test models with different hypotheses about perception in that respect.
- Inline with this remark, not all tasks included in the benchmark are behaviorally relevant for rodents. This might be why no clear conclusions can be deduced from the analysis of the task clusters. Another possible reason could be that some visual areas are responsible for detecting motion, which is why static stimuli do not allow for their differentiation from other areas.
- In the absence of a guiding hypothesis and as mentioned by the authors in the discussion, the benchmark should include a wider range of architecture types and objective functions. Especially biologically plausible models of vision, such as recurrent models. Objective functions include, but are not restricted, self-supervised objectives and generative models of vision.

Minor
- Calling a neural regression method state of the art is inaccurate and might cause confusion. A neural network architecture is state of the art if it performs better than previous architectures whereas the higher pearson's R on brainscore only highlights the new neural regression method's ability to extract more correlation from the data.
- The paper explores differences of neural regression on two axis; task and architecture. However, scores might change for different task-architecture combinations. It is definitely a formidable challenge to perform such an analysis at a large scale. But it is possible for a subset of architectures that do not share the same inductive bias (like convolutions), and using only behaviorally relevant tasks.
- Results in 3.6 which claim the slight importance of architecture over task are far from conclusive because many biologically relevant tasks are not included and the space of architectures is not fully covered as mentioned previously.

**Main Review:**

The paper presents a benchmark, new evaluation metrics, a large scale evaluation of vision models trained on various tasks as models rodent visual cortex, many analyses of the results and interesting interpretations of these results.

strengths:
- The benchmark is extensive, it explores a large number of models and tasks when compared to previous work in the field.
- They propose new metrics for neural regression which are faster and computationally inexpensive. Furthermore, the two metrics evaluate different aspects of the neural code (fitting individual neuron activity and fitting neural populations).
- Previous results on randomly initialized models are replicated and extended to other architectures. Which offers a new perspective on the inductive bias contributes to the scores.
- The layer-by-layer and area-by-area analyses offer many interesting hypotheses and predictions on rodent visual cortex.
- The paper is clear, thorough, well detailed and well explained. The code is provided for reproducing the results.
- The authors discuss the paper's limitations, provide detailed descriptions of the computational resources they used and acknowledge the ethical concerns behind their work.


**Time Spent Reviewing:**

12

---

> ### Author Response · Authors · 2021-08-10
> **Response to Reviewer ezgN**
>
> We thank the reviewer for their clear, comprehensive and thought-provoking feedback. We address limitations point by point below.
>
> Limitations (Major)
>
> - [Lack of Guiding Hypotheses] This is a duly noted, and deep point. As a largely data-driven analysis, we did not make strong theoretical assumptions about how the results would manifest. However, there were a few competing hypotheses and overarching questions we hoped our analysis would address and arbitrate. The first was the idea (implied by previous findings) that mouse visual cortex may effectively be a bank of random filters. This, in and of itself, is not entirely implausible -- research in olfactory cortex, for example (see [1] for an overview) has suggested ‘flexible random associations’ are the norm in some sensory systems, and a similar mechanism could be at play here. We specifically designed our benchmarking to arbitrate this hypothesis, and our data reject it strongly. A second set of claims (or debate) we hoped to address was the debate on whether or not mice use their visual systems for object recognition. This was precisely what the taskonomy results were meant to address -- with the idea being that if object recognition is one of the more predictive tasks (as it is, for example, in primate ventral stream), these kind of features must have some utility. This turned out not necessarily to be the case, but that, too, offers some potential insight. A third debate we designed our benchmarking to address was the question of whether (like primate visual cortex), mouse visual cortex shows an information processing hierarchy (as suggested by recent physiological studies); here our findings complement recent physiological studies nicely [3] by roughly recapitulating the proposed hierarchy of cortical areas from mouse V1 onwards (see also general note, at the end of this comment). Needless to say, what we’ve recounted here was not necessarily phrased as such in the paper, and we take to heart that we could highlight theoretical claims about mouse brain more emphatically in our paper.
>
> - x2 [Disjunct with Behavior / Alternate Modeling Regimes] Another excellent point. The mice (untrained adults) were indeed free viewing, and in that sense, none of our models constitute behavioral or developmental models -- only neural / representational models whose predictive power is contingent on how similarly they process the statistics of natural images. Mice may not be doing ‘object recognition’ or ‘edge detection’ per se, but their behavior may be contingent on the same kinds of representational associations, hierarchies or invariances learned through any number of computer vision tasks. Moving towards self-supervision is a high priority for us, and in that regard, we’ve since run a number of self-supervision models (e.g. from FaceBook’s Dino -- link below). While we believe these are a start, the key is in finding models that contrast different data augmentation schemes (as in SimCLR’s augmentation knockout) and highlight the efficacy of more biologically plausible transforms (gaussian / peripheral blur to simulate lower visual acuity, for example).
>
> - [Behavioral Relevance / Motion] While it’s true that the mapping between some of the computer vision tasks and behavior is tenuous (it seems very unlikely, for example) that jigsaw and colorization are relevant), others are behaviorally relevant and span early visual tasks (like edge detection) and later visual tasks (like segmentation). At best, the results are statistically conclusive in that they suggest no differentiation between cortical area for all tasks -- or at least, as detectable by taskonomy representations. Motion and temporality more generally (as the reviewer aptly points out) may be the key to further differentiation -- as suggested, for example, by [2], who claim optic flow can be used to differentiate areas VISal and VISpm. We will include this as a note in the appendix, and discuss candidate models for use with the Allen Observatory’s Natural Movies dataset. We entirely agree with the expansion to further kinds of models. Results from the self-supervised models we’ve run have been added as a table in our anonymous Github repository.
>
> Limitations (Minor)
>
> - This is a point very well taken, and one mentioned by other reviewers. As an example of a phrasing adjustment we intend to make, consider the sentence beginning at line 18 (‘we introduce a novel, highly optimized….’), which we will change to: “We introduce a more efficient, more accurate neural regression pipeline that achieves substantially higher scores than previous methods on the publicly available benchmarks of primate BrainScore.”
> - The reviewer is entirely right on this point. There are a few instances in which models share the same task (e.g. object recognition), but differ in their inductive biases (transformers versus convolutional architectures). We can point to two instances of this occurring (between Dino-ResNet50 and Dino-ViT-B for a self-supervised example; and between any transformer and convolutional architecture trained on Imagenet). More broadly, though, matching architectures between models that don’t share inductive biases seems a harder challenge, since their respective capacities have yet to be systematically compared. Even harder of a challenge is subsequently finding matched models with behaviorally relevant tasks suited specifically to mice (rather than human and nonhuman primates). If these models exist in open-source, and the reviewer is aware of them, we would be more than happy to test them during the discussion period. The addition of such models would be a great improvement!
> - This is a point other reviewers have made as well. We can soften the claims made here, adding the caveat that we’ve tested only a limited range of combinations so far. Given the miniscule numerical difference, we could also remove the suggestion of advantage entirely.
>
> Addenda
>
> - [Self-Supervised Models] The link to the scores of the self-supervised models we’ve tested so far is here (in our anonymous Github repository). We haven’t yet had the chance to analyze trends in their performance.
> https://github.com/annonymousecode/Model-Zoology-and-Neural-Taskonomy/blob/master/rebuttal_figures/self_supervised_model_scores.csv
>
> - General note: We somewhat haphazardly mentioned the notion of “hierarchy” in fig. 3 and sec 3.8 without providing references or statistics. We have attempted to rectify this with a figure we’ve added to our anonymous Github repository that shows pairwise comparisons and omnibus statistics in terms of the average depth of model layer that best predicts each visual cortical area. This figure shows that both our metrics (but especially the RSA metric) recapitulates a hypothesized hierarchy across area in mouse visual cortex, with primary visual cortex projecting to VISl in the ventral stream and VISal - VISrl (then, VISpm - VISam) in the dorsal stream. That figure may be found at this link:
> https://github.com/annonymousecode/Model-Zoology-and-Neural-Taskonomy/blob/master/rebuttal_figures/heirarchy_plots.png
>
> References
>
> [1] Kay LM. Olfactory coding: random scents make sense. Current Biology. 2011 Nov 22;21(22):R928-9.  https://www.cell.com/current-biology/pdf/S0960-9822(11)01132-8.pdf
> [2]  Andermann ML, Kerlin AM, Roumis DK, Glickfeld LL, Reid RC. Functional specialization of mouse higher visual cortical areas. Neuron. 2011 Dec 22;72(6):1025-39. https://www.sciencedirect.com/science/article/pii/S0896627311010129?via%3Dihub
> [3] D’Souza RD, Wang Q, Ji W, Meier AM, Kennedy H, Knoblauch K, Burkhalter A. Canonical and noncanonical features of the mouse visual cortical hierarchy. bioRxiv. 2020 Jan 1. https://www.biorxiv.org/content/10.1101/2020.03.30.016303v1.full

---

> > ### Comment · Reviewer_ezgN · 2021-09-17
> > **Score Increase**
> >
> > I would like to apologize for the late response and thank the authors for the thorough response.
> >
> > -   [Lack of Guiding Hypotheses] I fully understand the reasoning behind proposing a large scale data-driven analysis. It allows for answering many questions on the brain. My review is pushing the authors on the accuracy of the answers provided by this large-scale analysis and whether they take into account confounds. Nonetheless, the paper provides a good starting point for further experiments disambiguating some of the proposed answers (on mice object recognition and mice visual system hierarchy for example).
> >
> > - [Disjunct with Behavior / Alternate Modeling Regimes] I acknowledge the point of using neural networks purely as neural representation models, although I believe taking the behavior into account when comparing architectures would make more sense. More concretely, would the ranking in table 1 change if models were trained on objective functions closer to "free viewing" rather than object recognition ? (even though it would be time and resource consuming to answer this question since it implies retraining all models included in the analysis with a different "free viewing" objective). I believe moving towards self-supervision, as the authors did, might be a good step in that direction.
> >
> > -  [Behavioral Relevance / Motion] The taskonomy analysis does include behaviourally relevant tasks. My criticism concerns the irrelevant ones. In my opinion, the benchmark already provides answers to some of the questions addressed in the paper. However, it could simpler and more extensive by:
> >     - exploring more inductive biases (like top down connections or recurrence) and choosing one representative model per inductive bias. This would reduce the number of CNNs included in the analysis for example.
> >     - focusing only on behaviorally relevant tasks and excluding other ones.
> >
> >     Including more inductive biases would bring more insights (especially when they include biologically plausible ones).
> >
> > Overall, I find that the paper benefited from the additions made by the authors in response to all reviews and I improved my score accordingly. I trust the authors will take the proposed changes into account in the final submission.

---

### Author Response · Authors · 2021-08-30
**Summarizing Responses to Feedback**

As the reviewing period comes to a close, we wanted to again thank all reviewers for their extensive and insightful comments. We believe adjustments that we made, including several new analyses and clarifications requested by the reviewers, will render the manuscript much stronger. To facilitate the final overall review, we’ve summarized the main adjustments we’ve made below:

- Figures: In response to reviewer omuc, we made a number of new figures to better highlight key findings, all of which may be found in our anonymous Github repo (https://github.com/annonymousecode/Model-Zoology-and-Neural-Taskonomy). Particularly compelling, we think, is a figure we made to show the hierarchy of information processing from VISp onwards, a figure we deliberately designed for easy comparison with figures elsewhere in the literature. That figure may be found here:
https://github.com/annonymousecode/Model-Zoology-and-Neural-Taskonomy/blob/master/rebuttal_figures/heirarchy_plots.png
- Linking to Literature: We added a number of references to our discussions on how the work recapitulates findings from previous studies of rodent visual cortical (functional) neuroanatomy, in particular discussions of functional differentiation and processing hierarchies.
- New Models Tested: In response to reviewer ezgN, we tested a number of self-supervised models, showing in some cases a parity of predictive performance with supervised models. Those results may be found in our anonymous Github repository.
- Motivation: A number of reviewers asked for a bit more theoretical motivation than we originally provided in the submitted draft of the manuscript. Our main responses (at least a few of which we hope to include in the final manuscript) were:
     - That the extension of deep neural network modeling to mouse visual cortex provides an important neuroscientific foil to primate visual cortex, probing potential points of evolutionary divergence and in the ideal clarifying what role vision may serve in each of these species, as well as how these systems are organized.
     - That comparing trained and randomly initialized models actually provides a relatively robust refutation of the suggestion that mouse visual cortex may be organized like other sensory systems that rely on “flexible random associations”.
     - That using deep neural networks to uncover the task structure of mouse visual cortex may be key to moving beyond object recognition as the sole paradigmatic task of biological vision, and facilitating the exploration of new functionalities that could in turn inform new modes of training machine vision models.

We still would welcome responses from those reviewers that haven’t had a chance yet to respond, and hope that those who were on the border of accepting are sufficiently convinced by changes made. Thank you!

---

### Decision · Program_Chairs · 2021-09-27

**Decision:**

Accept (Poster)

**Comment:**

This paper presents a large-scale / systematic study fitting many modern deep nets to mouse calcium data from the Allen institute. This paper received two accepts, one borderline accept and one borderline reject and was discussed on the forum. There was a bit of a discussion regarding the actual contributions of this paper. There was some agreement between the reviewers that the authors oversold the claim that they introduced a novel regression methods and the authors agreed to remove or significantly downplay this claim.There was also some discussion about the lack of clear hypotheses being tested or significant results contributing to our understanding of the visual cortex. The study is an impressive feat from the engineering perspective but at the end of the day from the dozens of models and results presented the fit are relatively low (correlation coefficients r<0.2 which means r^2<0.04). This is to be expected given how noisy calcium data are and given the small number of stimuli available (~100). However, given these scores it is possible that we are essentially fitting noise and indeed  the models are all performing very similarly so there is no really salient results. With that said, the positive reviewers also emphasized that the present study seem to be at least partially addressing a controversy in the field (because a previous study had found that random models did about as well as trained ones). Here at least the authors report the accuracy of trained models that significantly outperform untrained ones. The code is also made freely available and positive reviewers see that as a "launching point" for additional work on mouse (as opposed to primate cortex). All in all, this really is a borderline submission. Given the issues highlighted above, this paper could be accepted if space permits.